# Towards Marine Dual Fuel Engines Digital Twins—Integrated Modelling of Thermodynamic Processes and Control System Functions

**Sokratis Stoumpos, Gerasimos Theotokatos \*[ID], Christoforos Mavrelos and Evangelos Boulougouris**

Maritime Safety Research Centre, Department of Naval Architecture, Ocean and Marine Engineering, University of Strathclyde, 100 Montrose Street, Glasgow G4 0LZ, Scotland, UK

**\*** Correspondence: gerasimos.theotokatos@strath.ac.uk; Tel.: +44 (0)-141-548-3462

**Abstract:** This study aims at developing an integrated model that combines detailed engine thermodynamic modelling and the control system functional modelling paving the way towards the development of high-fidelity digital twins. To sufficiently represent the combustion process, a multi-Wiebe function approach was employed, whereas a database for storing the combustion model parameters was developed. The developed model was employed for the systematic investigation of a marine four-stroke dual fuel engine response during demanding transient operation with mode switching and load changes. The derived results were analysed to identify the critical engine components and their effect on the engine operational limitations. The results demonstrate that the developed model can sufficiently represent the engine and its subsystems/components behaviour and effectively capture the engine control system's functionality. The appropriate turbocharger matching along with the sufficient design of the exhaust gas waste gate valve and fuel control systems are crucial for ensuring the smooth engine operation of dual fuel engines.

**Keywords:** marine dual fuel engines; 0D/1D simulation; control system functional modelling; operating modes switching; GT-ISE modelling; digital twins

## 1. Introduction

As the reduction of greenhouse and non-greenhouse emissions is amongst the high priority issues that the shipping industry has to endure on account of the stricter environmental standards imposed [1], the use of alternative fuels including natural gas, methanol and bio-fuels have been proposed for improving the environmental sustainability of maritime operations [2]. In specific, natural gas (NG) as a fuel has proven to be a viable solution for vessels operating both inside and outside the Emission Control Areas (ECAs) owing to the rapid development of the global liquefied natural gas (LNG) infrastructure [3], the lower LNG fuel price levels [4] compared to other fossil fuels [5] as well as the clean nature of lean combustion [6], which leads to the reduction of nitrogen oxides (NOx), carbon dioxide ($CO_2$) due to the low carbon to hydrogen ratio and the almost complete elimination of particulate matters (PM) and sulphur oxides (SOx) emissions [3].

The economic and environmental benefits of using LNG as an alternative fuel led the marine engine manufacturers to develop dual fuel (DF) versions of both the large two-stoke slow-speed engines and the four-stroke engines [7]. The marine DF engines run with a small amount of pilot diesel fuel used for initiating combustion and natural gas as the main fuel; they can operate according to either the premixed combustion concept (with the natural gas injected in the inlet ports during the cylinder induction phase [8,9] or within the cylinders during the compression phase [10]) or the gas

direct injection concept with the natural gas injected after the pilot fuel injection [11]. Other engine types include the spark ignited pure gas engines as well the gas–diesel engines [12]; the latter can operate with a variety of gas to diesel fuel fractions. The marine DF engines can operate either in the diesel mode by burning diesel fuel—heavy fuel oil (HFO) or marine gas oil (MGO)—or in the gas mode by burning natural gas and pilot diesel fuel (for initiating combustion); thus providing additional fuel flexibility for the vessel operation. In this respect, they are currently becoming the industry standard not only for LNG carriers but also for all LNG-fuelled vessels, as reported in [13,14].

In this respect, the continuous engine development and optimisation procedures, which are usually based on a number of techniques including experimentation, design, prototyping and engine mathematical modelling, are essential to the marine industry [12]. The advent of Internet of Things (IoT) and digitalisation has enabled the effective application of the digital twins in various industries including shipping, with expected benefits on the systems safety, maintenance, efficiency and environmental footprint [15]. The use of marine engines thermodynamic models as digital twins is reported in [16]. In this respect, high-fidelity integrated models combining the engine thermodynamic processes modelling with the control and safety system modelling are required as a first step prior to developing an intelligent digital twin as defined in [15].

As reported in [17,18], very few studies have been published focusing on the modelling/simulation or experimental analysis of marine DF engines. The more recent authors' studies [19,20] presented the transient simulation of a large marine DF engine and the derived results were employed for the safety analysis of the engine operations. In addition, to the best of authors' knowledge, other simulation studies on the transient operation of marine DF engines are not available in the open literature.

In [21], a marine DF engine was studied experimentally at steady state conditions complemented by a liquid fuel injection model to obtain a better understanding of the local combustion conditions. In Li et al. [22], a small marine DF four-stroke engine was experimentally investigated for comparison with the engine performance/emissions characteristics in the diesel and the gas operating modes. Boeckhoff et al. [23] presented the experimental investigation of a large marine DF four-stroke engine of the premixed combustion type at both steady state conditions (studying the effect of the engine and fuel parameters variations on the engine performance and emissions) and transient conditions (with fuel switching), reporting the engine operational experience. Banck et al. [24] conducted an experimental analysis of a large DF four-stroke engine optimised for marine applications discussing the engine load acceptance at various load ramp slopes and the engine operational requirements at the low load range. Portin [25] reported the development of DF four-stroke engines for offshore applications and presented experimental data for the engine load acceptance test from 40% to 80% load for the gas mode. In Mohr et al. [26], the experimental investigation of a large size four-stroke single cylinder engine at steady state conditions was reported and the influence of the engine settings variation on the engine performance and emissions was discussed.

Previous investigations reported that DF engines have operational limitations in marine and offshore applications due to the expected considerable transient loading requirements. In particular, during the load increase phase of premixed combustion DF engines running in the gas mode, the engine delivered power initially increases due to the injection of an additional amount of gas fuel, whilst the amount of combustion air increases more slowly due to the turbocharger lag effect. Thus, the engine runs temporarily with a rich gas fuel−air mixture until the turbocharger delivers the required air flow. In principle, the same takes place in a diesel engine, but due to the different combustion process, the load acceptance is subject to different limitations. Thus, over-fuelling a gas engine leads to a rich combustion (i.e., lower air−fuel ratio), which at low loads improves the engine combustion efficiency [25], whereas at higher loads, over-fuelling causes the engine operating point to shift towards the knocking limit, as the range between the misfiring and knocking borderlines becomes narrower at higher engine loads [21]. In contrast, over-fuelling a diesel engine at low loads reduces the combustion efficiency and leads to black smoke, if not controlled. However, at higher loads, the turbocharger's faster response improves the diesel engine load acceptance.

Considering the size, complexity and cost of the marine DF engines, experimental studies require significant resources. Thus, engine modelling as well as simulation is employed as one of the most effective methods for obtaining a better understanding of the engine operation and components interactions as well as predicting the engine performance and emission characteristics. In Stoumpos et al. [18], a large marine DF four-stroke engine steady state model was developed in the GT-ISE™ software and used for the parametric investigation of the engine settings and the study of the engine performance/emissions trade-offs. The employed models were of the zero-dimensional (0D) type for modelling the engine cylinders processes and the one-dimensional (1D) type for modelling the intake and exhaust manifolds. Theotokatos et al. [20] reported the extension of the model presented in [18] and simulated critical situations of the engine operation considering normal and delayed exhaust gas waste gate valve control providing recommendations for enhancing the engine safety. In Mavrelos and Theotokatos [17], a large marine DF two-stroke engine of the premixed combustion type was investigated based on a steady state 0D model developed in the GT-ISE software and the parametric optimisation of the engine settings was performed with the aim of simultaneously reducing the $CO_2$ and NOx emissions. In Ritzke et al. [27], a combined 0D/1D and computational fluid dynamics (CFD) three-dimensional (3D) approach was proposed for modelling a four-stroke dual fuel marine engine in the AVL Boost and FIRE software tools. The 0D/1D engine model was used to generate information regarding the initial and boundary conditions, and subsequently these conditions were used for the 3D CFD model. In Sixel et al. [28], a physical model for modelling the premixed combustion process of marine DF engines along with its integration with the engine 0D/1D model developed in the GT-Power software was reported. The model was used for the investigation of two engines (a single cylinder engine and a large marine DF four-stroke engine) operation at steady state conditions considering both natural gas and methanol as the main fuel. Based on the comparison with experimental data, the developed model accuracy was considered adequate for allowing its usage at the design phase of dual fuel or gas-diesel engines. Wenig et al. [29] developed a quasi-dimensional phenomenological combustion model to calculate the burning rate of a lean premixed mixture marine DF two-stroke engine with a pre-chamber. Following a process for the model constants calibration based on a set of experimental data, it was concluded that the model provided sufficient accuracy for a wide range of engine operating conditions.

In addition, a limited number of studies investigating the transient operation (including load chances and fuel changes) of small vehicles or heavy-duty DF engines by using simulation tools were published. Xu et al. [30] developed a one-dimensional model of an automotive four-stroke dual fuel engine in the GT-ISE software to study and improve the engine transient response by optimising the engine fuel injection. This model was validated against a comprehensive set of experimental data, from which a lag in the engine power delivery under transient loading with dual fuel operation was identified, and subsequently was applied to generate the required insight and to design control strategies for smooth torque delivery under dynamic conditions. For modelling the combustion process, a triple Wiebe function was employed, the constants of which were correlated with the engine indicated mean effective pressure based on the acquired experimental data as reported in [31]. Barroso et al. [32] modelled a heavy-duty DF compression ignition (CI) engine by employing a 1D model in GT-Power; the model was calibrated by using engine mapping experimental data and used in both steady state and transient conditions. Georgescu et al. [33] investigated the transient behaviour of gas and DF engines running on natural gas by employing two mean value dual cycle models. The first model was used to gain insight into the DF engine in-cylinder combustion process, whilst the second was used to simulate the entire engine system. The simulation results were used to discuss the engine limitations and transient response. In Mayr et al. [34], the development and implementation of a methodology was presented for simulating the transient response of large gas engines on a single cylinder engine test bed. Simple and fast models and algorithms based on lookup tables were employed to provide the boundary of the investigated engine components conditions. The simple model results were compared with the results from a 1D model developed in the GT-Power software demonstrating sufficient accuracy.

For operating a marine DF engine in a broad envelope as well as accommodating the demands during the fuels switching, the engine control system design as well as sufficient control strategies development is required. Wang et al. [35] studied the design of a marine DF engine fuel control system to accommodate effective fuel transitions by employing a mean value model-based approach. It was concluded that a Multiple Input Single Output (MISO) control system architecture with feedback corrections applied to both the gas and diesel fuels is advantageous compared with architectures that apply corrections to only one fuel command (either diesel or gas). Schmid et al. [36] focused on the marine four-stroke gas and DF engines cylinder individual combustion control by employing an anti-knock approach, determining ways of combining combustion balancing (which is currently a challenging task due to the extreme sensitivity of the modern engines to the cylinder-to-cylinder variations) with conventional control functions. Ott et al. [37] investigated the cylinder combustion individual feedback control of a four-stroke DF engine. The centre of combustion and maximum pressure rise was controlled by actuating the start and duration of the pilot diesel fuel injection. Engine experimental analysis indicated that the proposed controller was able to compensate the influence of various disturbances. An applicable and comprehensive control strategy for an automotive natural gas/diesel engine was presented in [38]. Roecker et al. [39] demonstrated a method for controlling the diesel fuel injection in DF four-stroke engine in order to overcome the shortcomings of the natural gas port injection and improve the engine transient performance and implemented the developed control in two vehicles engines. Fathi et al. [40] discusses the homogeneous charge compression ignition (HCCI) engine control structure in order to effectively control these engines and obtain acceptable performance and emissions characteristics. However, the complete detailed modelling of the engine and its control system has not been reported in pertinent literature.

From the preceding literature review, the following research gaps were identified: (a) the lack of a detailed model to adequately represent the marine DF engines' behaviour along with their control system functionalities; (b) the need for the detailed engine control system description and modelling; (c) the lack of a thorough investigation of the marine DF engine processes during transient operations including modes switching; and (d) the need for mapping the engine components' limitations and the control system requirements for ensuring reliable and smooth engine operation during transients with load changes and modes switching.

For addressing these gaps, the present study aims at systematically investigating a marine DF four-stroke engine response by developing a detailed model for both the engine and its control system. The developed model is an extension of the versions presented in [18,20] for simulating the engine steady state and transient operation, respectively. The focus of the present study is on the analysis of the engine response and the engine subsystems/components interactions as well as the identification of the engine operational limitations.

The model was developed in the GT-ISE software [41], as this software provides the tools, libraries and functionalities to address the inherent complexity of the engine and its control system modelling as well as the interfaces required for the programming of the controller logical functions. A similar implementation by using an in-house software would be more time-consuming. In addition, GT-ISE is a tool that has been extensively used in both academia and industry for modelling a considerable variety of engine types/sizes and fuels. Following validation under steady state and transient conditions, the developed model was used for a systematic analysis of the engine response under the investigated transient operating conditions. By comparing the engine performance parameters metrics, the critical engine components are identified, and the engine operational limitations are discussed, which provided valuable information for the understanding of the engine response and the control system design improvement.

The original contribution of this study is summarised as follows: (a) it is the first time that the detailed DF engine control system functional analysis and modelling is presented in pertinent literature; (b) the development and extensive validation of the complete engine and its control system model, which allows for simulation of the engine performance/emission and response at both steady state and transient operation including modes switching; (c) the investigation of a large four-stroke marine DF

engine transient operation in detail, enabling greater understanding of the involved processes and the interactions between the engine components; (d) the study of the control system response effects on the engine components' operation; and (e) the identification of the engine operational limitations.

## 2. Investigated Engine Description

In the present study, the four-stroke, non-reversible, turbocharged and intercooled Wärtsilä 9L50DF engine was investigated [42]. The engine is capable of operating in two distinct modes, in specific: (a) the gas mode running on natural gas and light fuel oil (LFO) (which is used as pilot fuel for initiating combustion) and (b) the diesel mode, in which either heavy fuel oil (HFO) or LFO is used as the main fuel. The engine control system must be capable of smoothly switching between fuels during the engine operation. The engine's high-power output along with the fuel flexibility, low emissions, high efficiency and reliability renders this engine an attractive solution for both electric power generation and ship propulsion [43]. Its main advantage is the lean-burn combustion, which provides an increased engine efficiency with reduced in-cylinder peak temperatures, thus resulting in reduced NOx emissions and engine thermal loading.

In the gas mode, the investigated DF engine operates at a much lower NOx emission level (compared to diesel mode) complying with the IMO Tier III limits. However, to achieve stable combustion conditions, an air−fuel ratio operating window between the limits of misfiring and knocking combustion needs to be targeted. The engine cylinders air−fuel ratio is adjusted via an electronically controlled exhaust gas waste gate (WG), which bypasses a part of the exhaust gas along the turbocharger (TC) turbine [42].

Each engine cylinder is equipped with a combined diesel and pilot fuel injector. Gas is injected by using solenoid gas admission valves at each cylinder inlet port (upstream the intake valves) during the engine induction process. The gas admission valves as well as the diesel fuel injectors are electronically controlled (in the gas and diesel operating modes, respectively) to adjust the engine power output in order to keep the ordered engine speed. The injected pilot fuel amount depends on the engine operating mode and load. The engine advanced automation system controls the engine functions, counting for the prevailing ambient conditions and the used fuel properties (including fuel quality, methane number, etc.), so that optimal running conditions are obtained [25].

In this study, the examined engine was considered as a part of a generator set operating at a constant speed of 514 r/min. The engine details are reported in the manufacturer product guide [42], the main engine characteristics are illustrated in Table 1, whilst the engine layout and components are presented in Figure 1. To allow for the engine's smooth operation, an adequate control strategy for governing the injected fuels amount as well as their injection timing and duration has to be implemented. This also includes the waste gate control, which is essential to adjust the combustion air−fuel equivalence ratio (λ) by regulating the boost pressure, when the engine operates in the gas mode.

**Table 1.** Main engine characteristics.

| MCR Power/Speed | kW/r/min | 8775/514 |
|---|---|---|
| BSEC at MCR (gas mode) | kJ/kWh | 7300 |
| BSFC at MCR (Diesel mode) | g/kWh | 190 |
| No. of cylinders | - | 9 |
| Bore/Stroke | mm | 500/580 |
| Turbocharger units | - | 1 |

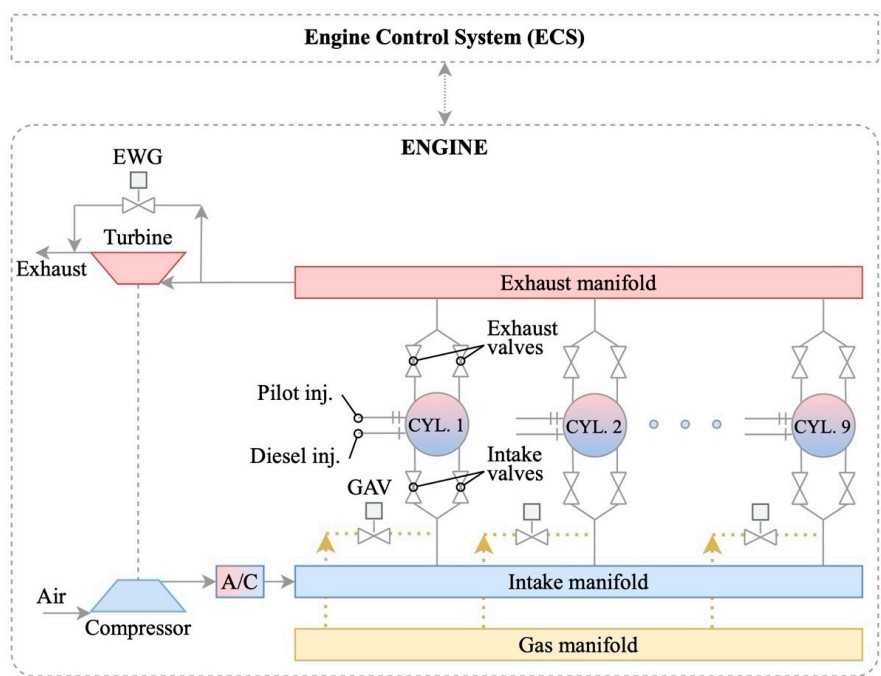

**Figure 1.** Wärtsilä 9L50DF engine layout.

*Engine Transient Operation Requirements*

For ensuring the engine integrity and smooth running during transient operations, the ECS needs to satisfy the engine response requirements as determined by the engine manufacturer [42]. With regard to the load transition, for load steps in the gas mode, the maximum acceptable step load increase is as illustrated in Figure 2 (left), whilst the maximum allowed step load decrease should be according to the following schedule: 100–75–45–0% (for the intermediate engine loads, the nearest lower load threshold needs to be used). In addition, the recovery time (i.e., the time required for the engine to reach its steady state operating point following a transient) should be less than 10 s, whilst the recommended time between consecutive load steps should be greater than 30 s. In the diesel mode, the maximum acceptable step load increase is as indicated in Figure 2 (left) and there are no limitations in terms of step load reductions. In this case, the recovery time after a load change decreases to 5 s, whereas the recommended time between consecutive load steps is greater than 10 s.

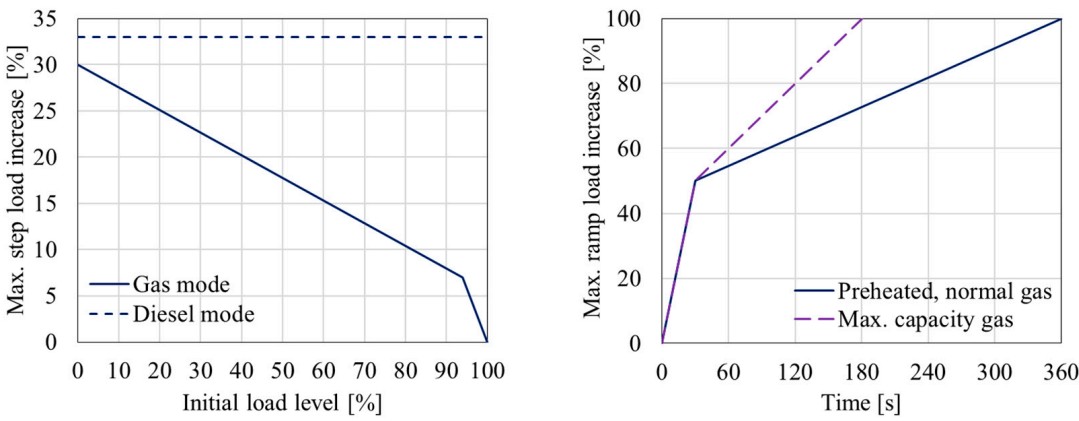

**Figure 2.** Maximum allowed step load increase in percentage of MCR for the gas mode and the diesel mode (left); Maximum allowed ramp load increase for engine operating at nominal speed (right).

For ramp load changes in the gas mode, the engine control system must not permit a load reduction from 100% to 0% faster than 20 s prior to automatic transfer to the diesel mode. The maximum allowed ramp load increase for various engine operating conditions is shown in Figure 2 (right). The curve "preheated, normal gas" is used as the default ramp load increase for the gas mode, whereas the curve "max capacity gas" indicates the maximum allowed ramp load increase.

Furthermore, considering the fuel change at any fixed load, the gas to diesel (GTD) mode switching needs to take place at any load within 1 s. The switch from the diesel mode to the gas mode (DTG) needs to be completed within 2 min for the minimization of disturbances to the gas fuel supply systems. In both cases, the maximum allowed speed drop is 10%. These manufacturer transient requirements are summarised in Table 2.

**Table 2.** Engine transient response requirements [42].

| Load Change | Diesel Mode | Gas Mode |
|---|---|---|
| Recovery time | ≤5 s | ≤10 s |
| Time between load steps | ≥10 s | ≥30 s |
| Maximum allowed speed drop | 10% | 10% |
| Maximum step-wise load increase | 33% of MCR | shown in Figure 2 |
| Maximum step-wise load decrease | No limitation | 100–75–45–0% |
| **Mode switching** | **Diesel to Gas** | **Gas to Diesel** |
| Required time | 2 min | 1 s |
| Maximum allowed speed drop | 10% | 10% |

## 3. Investigated Engine and Controls System Modelling

The present study focuses on the modelling of the investigated engine and its control system for further study of the engine response at transient conditions by employing the GT-ISE software, which is a widely used simulation program for engine modelling and analysis [41]. The complete engine model was realised by using the following assemblies of the GT-ISE software: (a) the 0D/1D engine model; (b) the user input; (c) the engine control system (ECS); and (d) the engine monitors and alarms. In the developed model, the user can order a specific transient operation (i.e., load changes at either the gas or the diesel modes, a mode switching at constant load or extreme load changes that may result in a mode switching) by employing the user input assembly.

### 3.1. Engine Modelling

The 0D/1D engine model for the investigated engine simulation at steady state conditions (for both the gas and diesel operating modes in a number of operating points) was previously developed in the GT-ISE software and was used for the engine settings optimisation as described in [18]. The extension of the existing model to accommodate the engine transient conditions was realised by: (a) setting up the input data block for providing the ordered engine operating schedule in terms of mode and load versus time (i.e., no fuel change, GTD or DTG mode switching and/or load change); (b) modelling the engine control system by developing the ECS assembly (based on the available published manufacturer data [25,42] as described in Section 3.3) to control the engine mode switching as well as the injected fuels; and (c) extending the engine model assembly to accommodate the modelling of the combustion process in transient conditions (as described in the Section 3.2).

The complete engine layout of the model in GT-ISE is presented in Figure 3. The engine model uses a number of elements available in the GT-ISE software libraries, which are appropriately interconnected. In specific, cylinder elements are used, connected upstream and downstream the intake and exhaust valves, respectively, whereas pipes and junctions are used for modelling the inlet and exhaust manifolds. The turbocharger (TC) unit is modelled by using the compressor and turbine elements; the former is

connected between the ambient and the air cooler, whereas the latter is connected between the exhaust pipe and the exhaust ambient. The compressor and turbine elements are mechanically connected with the TC shaft element. An air cooler element is connected between the compressor and the inlet manifold pipes, whereas the waste gate (WG) valve element is connected in the exhaust manifold for bypassing the turbine and is controlled by the boost pressure (acquired from the inlet manifold connected downstream the air cooler). The gas admission valves (one per cylinder) are connected in the engine inlet port (upstream the engine intake valves). The gas injection takes place during the respective cylinder induction process after the exhaust valves closing, so that all the injected gas remains into the engine cylinders. The diesel fuel and the pilot fuel injectors are directly connected to the engine cylinders. The engine cylinders are mechanically connected to the engine crankshaft, which is also connected to the engine load.

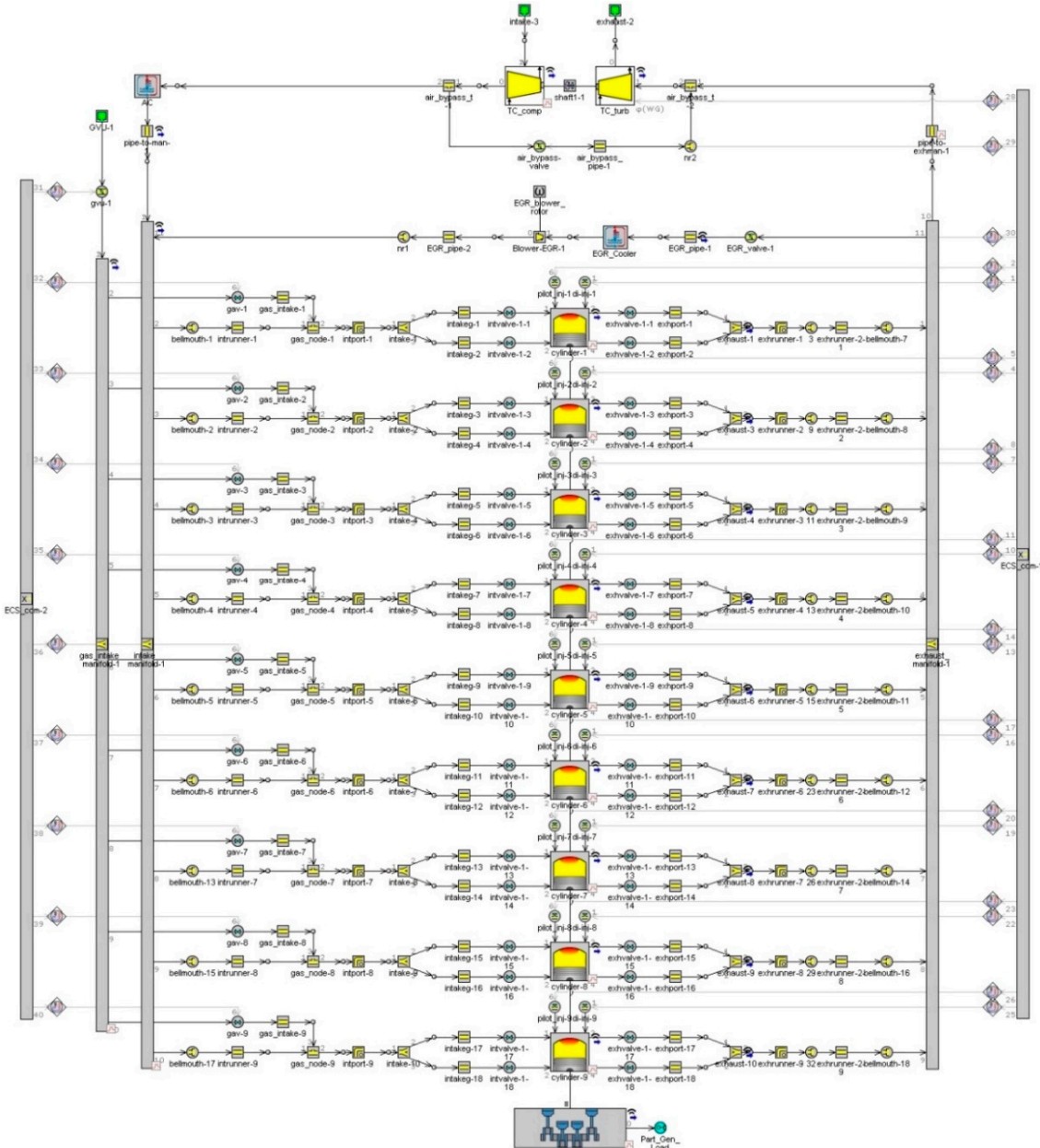

**Figure 3.** Engine model layout in the GT-ISE environment.

The engine cylinders are modelled by using a zero-dimensional method using a two-zone approach for modelling the combustion and expansion processes (one zone containing the combustion products and an unburned mixture zone), as well as a single zone approach for the remaining of the cycle [44]. The following mass and energy conservation equations were employed for modelling each zone along with the ideal gas equation and the cylinder volume time derivative equation:

$$\frac{dm}{dt} = \sum_i \dot{m} \tag{1}$$

$$\frac{d(me)}{dt} = -p\frac{dV}{dt} + \sum_i (\dot{m}H) - \dot{Q}_{ht} \tag{2}$$

where $m$ and $V$ denote the working medium mass and cylinder volume, respectively; $\dot{m}$ is the mass flow rate entering or exiting the cylinder; $p$ denotes the cylinder pressure; $e$ is the working medium total specific internal energy (internal energy plus kinetic energy per unit mass); $H$ is the working medium total specific enthalpy; and $\dot{Q}_{ht}$ is the heat flow rate from the gas to the cylinder walls.

According to the employed two-zone approach [41], the unburned gas zone consists of air and combustion products from the previous cycle, whereas the burned gas zone is generated after the start of combustion. At each time step, the amount of fuel and air is transferred from the unburned zone to the burned zone as dictated by the burning rate, which is calculated with the employed combustion model. The chemical kinetics calculation considering dissociation effects is carried out in the burned gas zone taking into account the used fuel(s) combustion and the assumption that the combustion products consist of the following 13 species: $N_2$, $O_2$, $H_2O$, $CO_2$, $CO$, $H_2$, $N$, $O$, $H$, $NO$, $OH$, $SO_2$ and $Ar$. The gas properties are calculated by using the species mass fractions and the respective property, the latter is calculated as algebraic functions of temperature.

Appropriate heat transfer, combustion and friction models [41] are employed. In particular, for calculating the gas to wall heat transfer coefficient, the Woschni heat transfer model is used [45]. The Chen–Flynn friction model is employed for calculating the engine friction mean effective pressure [46]. The cylinders volume is calculated by using the engine kinematic mechanism geometry. The employed combustion process modelling approach is described in the following section. For estimating the NOx emissions, the extended Zeldovich mechanism is employed, which is described in [47,48], taking into account the temperature of the burned gas zone. The model constants are calibrated for a discrete load at each operating mode.

A one-dimensional approach is used to model the pipes and junction elements by solving the following momentum conservation equation along with the mass and energy conservation equations described by Equations (1) and (2) in each discretised pipe element of the intake and exhaust manifolds [41]:

$$\frac{d\dot{m}}{dt} = \frac{Adp + \sum_i (\dot{m}u) - \left(\frac{4C_f dx}{D} + K_p\right)\left(\frac{1}{2}\rho u|u|A\right)}{dx} \tag{3}$$

where $\dot{m}$ is the boundary mass flux ($\dot{m} = \rho A u$); $\rho$ is the density; $A$ is the pipe cross-sectional flow area; $u$ denoted the velocity at the boundary; $C_f$ is the friction factor; $K_p$ is the pressure loss coefficient; $D$ is the pipe equivalent diameter; $dx$ is the discretization length; and $dp$ is the pressure differential acting across $dx$.

The pipe elements model solver employs an explicit time integration method, which provides a compromise between the required computational time and accuracy. The model variables include the working medium mass flow, density and internal energy. The pipe elements employed for representing the engine intake and exhaust manifolds are divided into a number of discrete elements considering a discretisation length of 0.4 to 0.55 times the cylinder bore diameter for the intake and exhaust manifolds, respectively. The scalar variables (pressure, temperature, density, internal energy, enthalpy, species

concentrations, etc.) are assumed to be uniform over each discrete element, whereas the vector variables (mass flux, velocity, mass fraction fluxes, etc.) are calculated for each discrete element boundary.

The compressor and turbine models use the steady state maps of the respective elements in a digitised format. The inlet and exhaust valves elements use the respective valves' profiles, employing the quasi-steady adiabatic flow equation for calculating the respective flow rates. The intake valves employ the Miller timing, closing before the cylinder bottom dead centre (BDC), which reduces the required compression work and the combustion temperature and results in higher engine efficiency and lower NOx emissions.

The angular momentum conservation equations are employed in the engine mechanical elements (shafts) in order to calculate the respective rotational speeds. Furthermore, the WG valve element employs a simplified PID controller with predefined proportional and integral constants for controlling the valve area considering the engine boost pressure as the input. This PID controller constant, which affects the WG valve response, is calibrated during the model calibration phase. The engine air cooler element is modelled by using multiple pipes connected in parallel that are treated according to the 1D model described above, where the heat transfer from the air to their walls is calculated considering the overall heat transfer coefficient. This heat transfer coefficient, as well as the heat transfer area and the cooling water temperature, are the model input parameters; the latter was considered constant for simulating both steady state and transient conditions.

The input data required to set up the engine model includes the engine geometric data, the cylinder valves profiles, the compressor and turbine maps, the WG valve area, the constants of engine sub-models (combustion, heat transfer and friction), the ambient conditions as well as the engine load and mode time variation. Initial conditions need to be provided for the temperature, pressure and composition of the working medium contained in the engine cylinders, pipes and receivers. The input data were acquired from the engine manufacturer product guide and the three-dimensional engine drawings were available from the engine manufacturer in [42].

*3.2. Combustion Process Modelling*

For calculating the fuel burning rate at the diesel operating mode, the single Wiebe combustion model is employed along with the Sitkey equation for estimating the ignition delay [44]. For modelling the combustion process at the gas operating mode, a triple Wiebe function is employed with each function representing the premixed combustion of a portion of the pilot fuel, the diffusive combustion of the remaining pilot fuel and the rapid burning of the gaseous fuel as well as the tail combustion of the cylinder residuals [6]. The ignition delay for the gas mode is approximated by using the equations and data reported in [28,49]. The gas admission valves are modelled by controlling the pulse width/duration (taking values in the region from 38 to 68 °CA from low to high loads) considering that the respective fuel pressure linearly varies with the engine load. The injected gaseous fuel mass flow rate is calculated as a function of the solenoid valve nozzle area, pressure ratio and the gaseous fuel properties upstream the gas admission valve. The main diesel fuel injected amount is modelled as a function of the engine load, whereas the pilot fuel amount is considered to be a function of the engine load and operating mode (diesel or gas).

The cumulative fuel burnt for the gas mode is calculated according to the following equation [41]:

$$x_{b,g}(\theta) = \sum_{i=1}^{3} \left[ FF_{g,i} \, x_{b,g,i}(\theta) \right] \tag{4}$$

where $i$ denotes the Wiebe function; *FF* denotes the weight of each Wiebe function ($\sum_{i=1}^{3} FF_{g,i} = 1$); and $\theta$ denotes the crank angle (top dead centre (TDC) of the closed cycle is at 0°CA).

Each individual Wiebe function is calculated by the following equation [44], which is also used for calculating the cumulative fuel burnt at the diesel operating mode:

$$x_{b,g,i}(\theta) = 1 - exp\left[-a\left(\frac{\theta - \theta_{SC\_i}}{\Delta\theta_{g,i}}\right)^{m_{g,i}+1}\right]$$  (5)

where $i$ denotes the Wiebe function; a is the Wiebe function parameter (considered 6.9); $\theta_{SC\_i}$ is the start of combustion; $\Delta\theta_{g,i}$ is the combustion duration; and $m_{g,i}$ denotes the i-th Wiebe function shape factor.

The combustion heat release rate is calculated using the derived fuel burning rate, which is the time derivative of the cumulative fuel burnt from Equation (4), and the total energy from all the injected fuels, according to the following equation:

$$\dot{Q}_b = \dot{x}_b E_{f,total} = \dot{x}_b \sum_{i=1}^{3} m_{f,i} LHV_i$$  (6)

where $\dot{x}_b$ denotes the fuel burning rate, $E_{f,total}$ is the total energy of all the injected fuels, $m_f$ is the burnt fuel amount, $LHV$ denotes the fuel lower heating value and $i$ denotes the fuel (gas, diesel, pilot). In this study, marine gas oil (MGO) was considered for the main and pilot fuels, whereas methane was considered for the gaseous fuel.

It was reported in the previous authors' studies [18,19] that the single Wiebe function can sufficiently capture the combustion processes in the diesel mode, as the maximum cylinder pressure, the brake specific fuel consumption (BSFC) and the indicated mean effective pressure (IMEP) were predicted with adequate accuracy. However, the triple Wiese function model was required to provide sufficient accuracy in the gas mode as reported in pertinent literature [18,19,30,31]. Hence, the Wiebe function modelling approach was also employed in this study, instead of a predictive combustion model, as the latter requires a considerable set of the model constants calibration [28,29].

The combustion model parameters (weights, start of combustion, combustion duration and shape factor for each Wiebe function), which determine the combustion profile for both the diesel and the gas modes, were calibrated at 25%, 50%, 75% and 100% loads (at steady state conditions), so that the predicted engine cylinder parameters (maximum pressure, IMEP and brake specific fuel/energy consumption) sufficiently matched their respective experimental values. The calibrated values of the combustion model parameters (controlled parameters) are stored in a database in the format of three-dimensional matrices (or dependency templates) as functions of the following two controlling parameters: (a) the engine load, and (b) the engine operating mode (diesel or gas).

The procedure illustrated in the flowchart of Figure 4 is employed for modelling the combustion process in each engine cylinder when the engine operates in transient conditions. The combustion model controlling parameters, which are taken from the ECS model, include the engine load and the operating mode (diesel, gas, GTD change, DTG change) as well as the fuels injected amounts (the values of fuels amount are used only for the case of the DTG change). For the engine operation in the diesel mode, the gas mode and the GTD change, quadratic interpolation is used (considering the engine load as the controlling parameter) for calculating the values of the respective combustion model parameters for each engine cylinder (single Wiebe function model for the diesel mode; triple-Wiebe function for the gas mode). It needs to be noted that during the GTD mode switching, the engine cylinders operate either in the gas mode for the cylinders in which combustion or injection processes already started prior to the implementation of the change, or the diesel mode for the cylinders in which combustion or injection processes started after the mode switching order (as in this case the gas fuel is cut off instantly and only diesel fuel is injected). Based on the combustion model parameters, the fuel burning rate and the heat release rate are calculated by using Equations (4) and (6), respectively. This calculation procedure is depicted in the interior box shown in the left-hand side in Figure 4.

The controlled combustion model parameters for each engine cylinder are updated at every cycle based on the controlling parameters derived from ECS described in 3.3.

The modelling of each cylinder combustion process for the DTG mode switching requires a more sophisticated approach as the engine cylinders operate for a considerable period (around 2 minutes) with the main diesel fuel, the gas fuel and the pilot fuel. The employed calculation procedure is described in the flowchart included in the exterior box in Figure 4. First, the total burning rates at the specific engine load are calculated considering separately the diesel mode and the gas mode by employing the procedure described in the previous paragraph (the procedure illustrated by interior box flowchart in Figure 4). In addition, the fuels amounts (derived from the ECS model) and the fuels lower heating values are used for calculating the fuels energy ratios for the diesel and gas modes according to the following equation:

$$ER_d = \frac{m_d LHV_d}{E_{f,total}}, ER_g = \frac{m_g LHV_g + m_p LHV_g}{E_{f,total}} \tag{7}$$

Subsequently, the total fuel burning rate is calculated by using Equation (8), which provides an adequate approximation of the total heat release rate for gas–diesel engines as deduced from the analysis of the experimental results reported in [50]. Finally, the heat release rate is calculated using Equation (6).

$$x_b = ER_d x_{b,d} + ER_g x_{b,g} \tag{8}$$

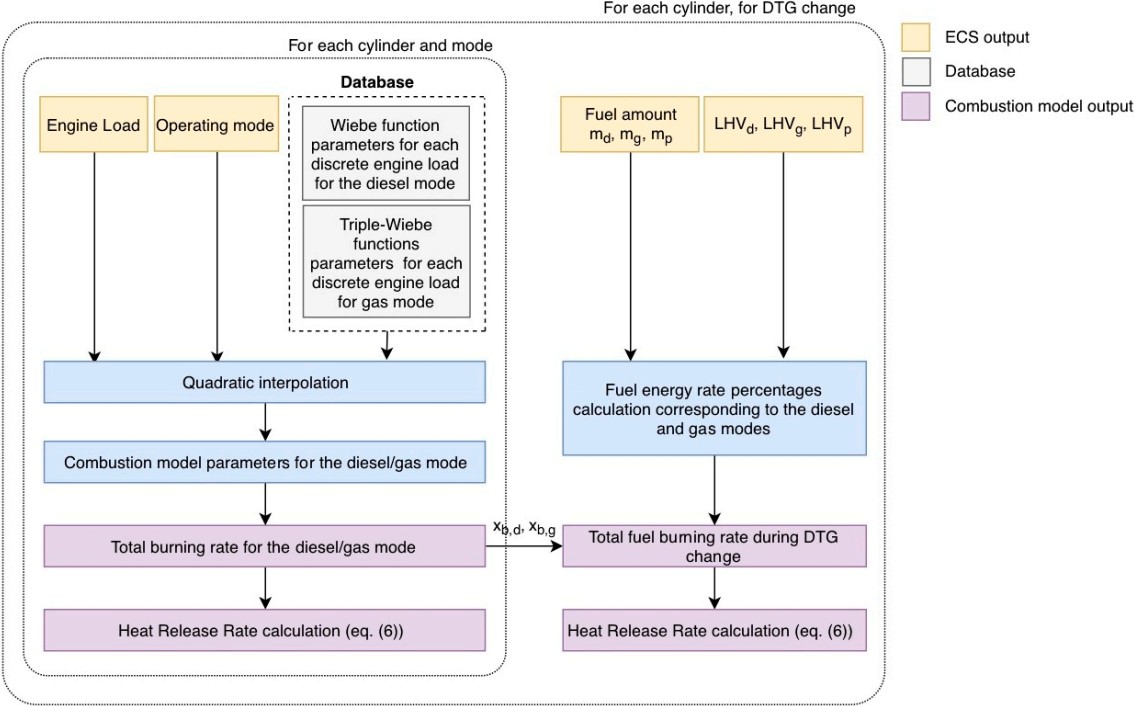

**Figure 4.** Combustion model procedure flowchart. Abbreviations: $m_d$: diesel fuel mass; $m_g$: gas fuel mass; $m_p$: pilot fuel mass; $x_b$: fuel burning rate, ECS: engine control system.

*3.3. Engine Control System Modelling*

The developed ECS model controls the gas admission valves, the diesel and pilot fuel injectors as well as the engine waste gate at both steady state and transient conditions with fuel or load changes. For the gas mode, the model is capable of identifying imposed step-wise load changes that exceed the maximum allowed load change (via comparison against the manufacturer maximum step load increase limitations), and subsequently implementing a mode switch from gas to diesel as specified by the engine manufacturer requirements. In such scenarios, the engine operation is immediately

switched to the diesel mode via a fast-acting signal, for the fuels and the waste gate controls. In addition, requests for a diesel to gas mode switching above 80% engine load are not allowed as indicated by the manufacturer [42]. The ECS operating mode controller was modelled to identify either the engine load or the fuel transition (based on the manual input for ordered operation mode) and accordingly calculate the respective combustion model parameters.

The developed ECS model assembly consists of the fuels (diesel main, diesel pilot and gas) and the exhaust waste gate controllers. A flowchart illustrating the modelling philosophy and the logical conditions of the developed ECS model is provided in Figure 5, whereas the structure and functionality of the developed fuel control system is illustrated by using the flow chart diagram presented in Figure 6. The ECS model includes the following elements: the logic controllers, the PID controllers for adjusting the injected fuels amount, the fuel transition profiles (i.e., GTD and DTG) as well as the exhaust waste gate (WG) controller. The ECS model employs two discrete control switches, which can be activated or deactivated though a logic controller (mode controller) based on the fuel or the fuels employed at the time (i.e., diesel or gas and pilot). In specific, for a load change at any operating mode, only one of the two switches is activated based on the operating mode. However, during the fuel transition operations, both switch controls are used in order to control the gas fuel, the pilot fuel and the diesel fuel injection timing. Additionally, the injection controllers for each engine cylinder were set to adjust the amount of all cylinder injected fuels and determine a suitable fuel change timing for each cylinder, based on each cylinder phase angle.

Upon an ordered operating mode, the developed fuel control system actuates the gas, the diesel and the pilot fuels injectors via a set of controllers, so that the engine is able to operate at: (a) steady state conditions in the diesel or the gas modes (i.e., fixed load and no mode switching); (b) load changes (transient operations) in the diesel or the gas modes; and (c) mode switching (transient operations) from the diesel mode to the gas mode and vice versa.

In particular, for the diesel mode simulation at steady state conditions, a PID controller (Diesel PID in Figure 5) adjusts the rack position of each cylinder diesel fuel pump that determines the fuel amount of the respective diesel fuel injector, based on the engine speed feedback signal. This controller employs a lambda limiter, thus preventing the engine to operate with low air–fuel equivalence ratio values, which may cause incomplete combustion issues and high thermal loading. According to the engine manufacturer [42], the pilot fuel is always injected in both engine operating modes, so that wear and damage of the pilot injectors are avoided. Hence, the pilot fuel injection control (Pilot controller in Figure 5) was set to appropriately adjust the pilot fuel amount in both the investigated operating modes.

For the gas mode, the gas fuel supply pressure is assumed to linearly change as function of the engine load, whilst an additional PID controller (Gas PID in Figure 5) adjusts the duration of the gas admission valve opening based on the engine speed feedback signal, thus adjusting the mass of the gas fuel injected per cylinder. The pilot fuel pressure is considered constant and the pilot fuel injection amount is controlled for each cylinder. Moreover, the rack position of the diesel fuel pumps is set to its minimum position (i.e., no diesel fuel is injected). It is expected that the engine speed error signal (i.e., difference between the ordered and the actual average engine speed) is zero during the engine steady state operation in either the diesel or the gas modes, and as a result the employed controllers set the corresponding controlling parameters. The employed PID controllers' settings were tuned by using the Ziegler–Nichols method according to the guidelines provided in [51].

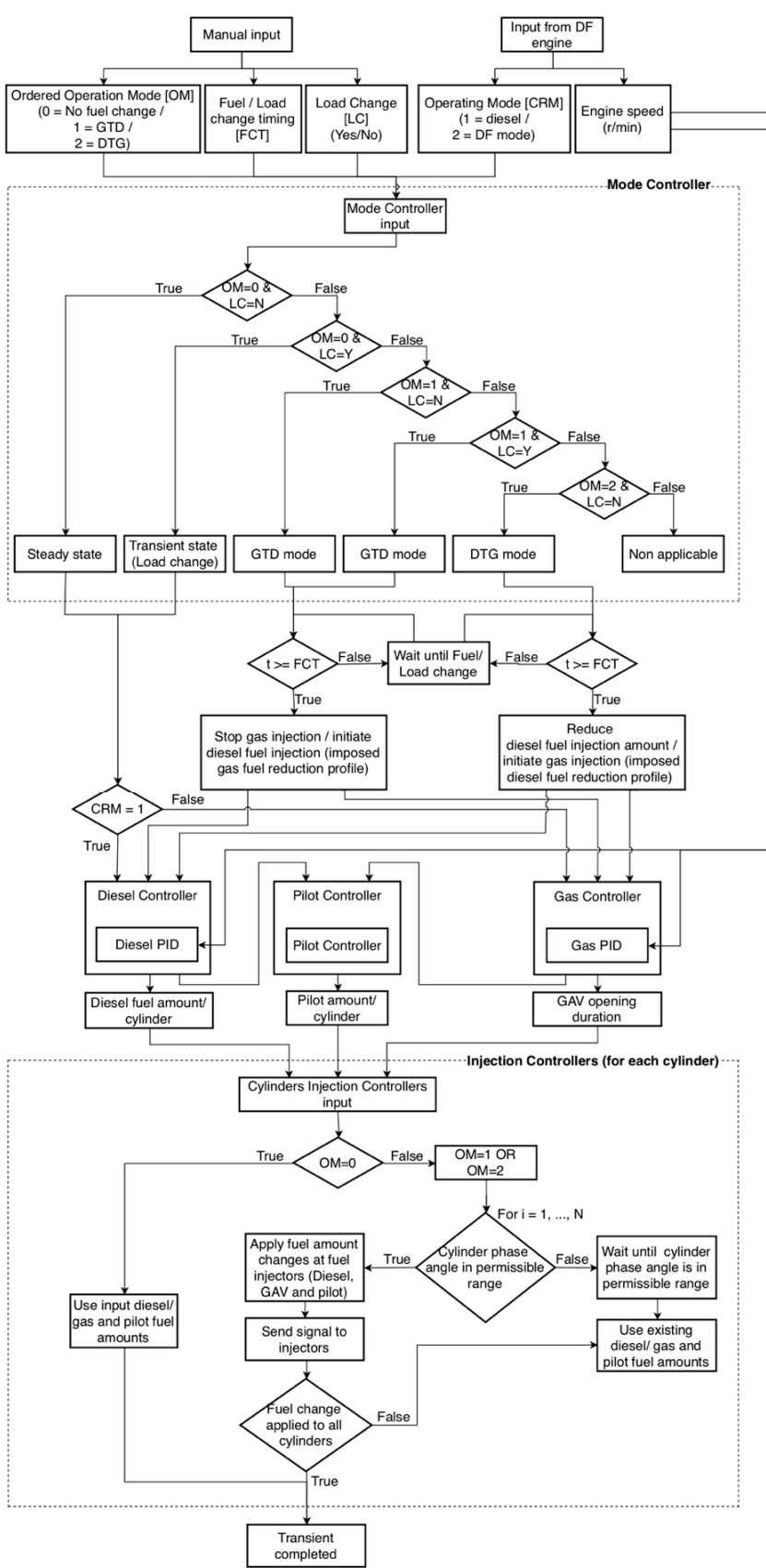

**Figure 5.** ECS model logical structure flowchart. Abbreviations: GAV: gas admission valve; GTD: gas to diesel; DTG: diesel to gas; CRM: Current Running Operating Mode.

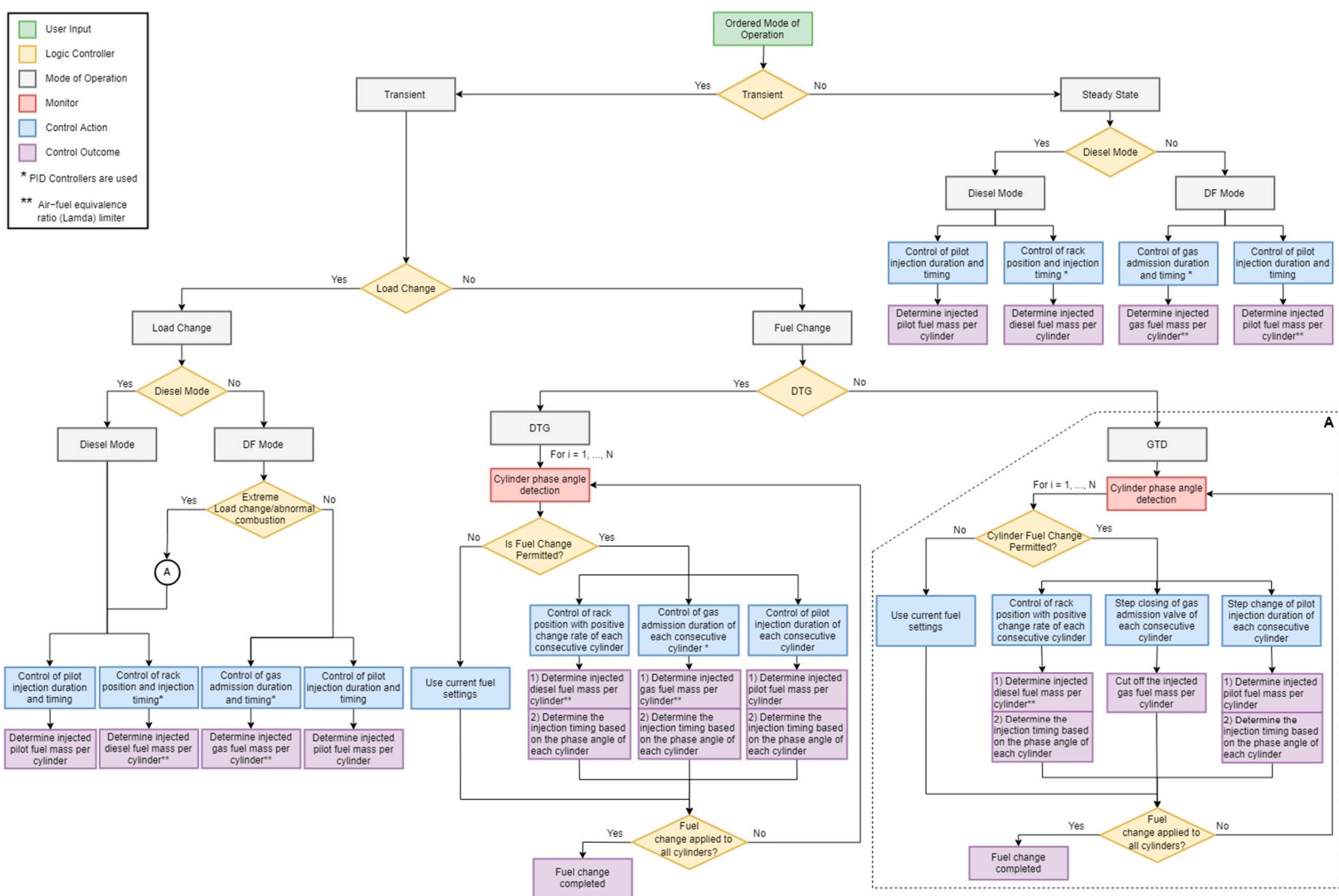

**Figure 6.** Fuel control system functional diagram.

For controlling the engine transient operation with load changes in the diesel or the gas modes, the fuel control system follows a similar control strategy as described above for the steady state operation. The controllers use the engine speed error (ordered speed to actual speed difference) as the input and eventually control the injected fuel amounts depending on the operating mode, aiming to minimise the engine speed error by adjusting either the rack position of each cylinder diesel fuel pump, or the gas admission and the pilot fuel injection duration. Additionally, the fuel control system is also designed to immediately change the engine operation to the diesel mode in cases where the ordered engine load change exceeds the investigated engine maximum load acceptance criteria when the engine operates in the gas mode. In this case, the transient operation involves both the load change and the mode switching from the gas to the diesel modes. The fuel control system performs all the necessary control actions (described in the following paragraphs) to initially achieve the mode switching (i.e., gas to diesel) and subsequently to respond to the ordered engine load change.

For the control of the engine transient operation with a mode switching from diesel to gas (DTG), the engine control system actually controls both the gas and diesel fuels (gas fuel pressure and injection duration and diesel rack position), whilst the pilot fuel injection duration is also adjusted to control the amount of the pilot fuel injected per cylinder. However, for the modelled engine control system, the amount of the gas fuel injected per cylinder is controlled via an imposed profile with a positive rate of change (slope) depending on the engine load, whilst the pilot fuel amount is calculated by interpolation, employing a lookup table with the measured values for each engine operating point. At the same time, the diesel PID controller adjusts the diesel fuel rack position based on the engine speed feedback signal, and thus, determines the mass of the diesel injected per cylinder and per cycle. It should be noted that the ECS model allows for the gas fuel injection only for the cylinders operating in their open cycle period.

In the case of an ordered gas to diesel mode switching (GTD), the rack position of each cylinder diesel fuel pump (determining the fuel amount of the respective diesel injector) is adjusted based on the feedback of the diesel PID controller. The gas admission valves of the engine cylinders, in which the injection has not started yet, are ordered to immediately close, based on an imposed step input signal, whilst a step change input signal governs the amount of pilot fuel injected in each cylinder.

In either the diesel or the gas modes, the fuel control system takes into account each cylinder phase angle during mode switching in order to determine if the ordered fuel change in each cylinder is permitted and, hence, define the timing for implementing this fuel change. This is considered in order to avoid a fuel change during the cylinders closed cycle or when the gas fuel injection is ongoing, which may lead to engine speed and power fluctuations. The fuel control system was set to perform the fuel change of each cylinder during the corresponding intake phase, prior to the gas fuel injection start. The completion of the ordered engine fuel change is achieved when the fuel change is implemented to all the engine cylinders. In this respect, the injection controllers were set to serve the purpose of identifying whether the fuel change is permitted in each consecutive fired cylinder and whether the fuel change has been applied to all cylinders. If the fuel change is not permitted in a cylinder due to its ongoing operating phase, the initial fuel settings are applied until it reaches the intake phase of the next cycle. The fuel change control actions are repeated until the fuel change is applied to all the engine cylinders [25].

It must be noted that in the ECS model, the crank angle (CA) position information was taken from the engine crankshaft block whilst considering the phase angle of each cylinder. The actual engine control system employs two CA sensors (encoders) for measuring the crank angle; the first is located in the flywheel, whereas the second is located in the free end.

## 4. Results and Discussion

Steady state runs were performed in a number of operating points, so that the derived performance and emission parameters are compared with the respective experimental data from the engine testbed trials. The steady state simulation results, along with a parametric investigation of the engine settings

effect on the performance–emissions trade-offs, are reported in an author's previous study [18]. From the data presented in [18], it can be inferred that the model accuracy is sufficient in all the investigated engine steady state operating points with the maximum value of the absolute percentage error being smaller than 3.5%.

Following the engine simulation at steady state conditions, the developed model in GT-ISE was used for simulating the engine transient operation including load and mode switching. Three cases, for which published experimental data are available, were investigated in specific:

(a)  Case 1—the engine operation at 100% load in the gas mode and a mode switch to the diesel mode [52];

(b)  Case 2—the engine operation at 80% load in the diesel mode and a mode switch to the gas mode [52]; and

(c)  Case 3—the engine operation at 40% load in the gas mode and a step-wise load increase to 80% engine load [25]. For this case, due to the large ordered load increase, a mode switch from the gas mode to the diesel mode also takes place [25].

The first two cases were also presented in the authors' previous study [20], where the validation of the model results is discussed. However, the focus of that study was the safety analysis of the engine systems, whereas this study focuses on analysing the engine response and the engine components interactions as well as identifying the engine operational limitations.

For the three investigated cases, the predicted variations of the engine parameters including the normalised rotational speed, the engine load and the normalised fuels amount (for the gas and diesel fuels), the engine boost pressure, the exhaust gas temperature before the TC turbine, the TC shaft speed, the waste gate opening, the air−fuel equivalence ratio (λ), as well as the locus of the TC compressor superimposed on the compressor map are presented in Figures 7–9, respectively. In these figures, the available experimentally measured parameter variations are also presented to serve the purpose of the developed model's validation. All the parameters except for the exhaust gas WG valve opening were normalised by using their corresponding values at 100% load for the diesel mode, whereas the WG valve opening was normalised by using its maximum area. It can be inferred from the presented results that the model can predict the engine parameter responses with an adequate accuracy as discussed in detail in the following paragraphs.

*4.1. Case 1—100% Load-GTD Mode Switching*

For the first investigated case, the engine parameter variations are sufficiently predicted as illustrated in Figure 7. Both the simulation and experimental results show that the change of the engine operating mode from gas to diesel took place within 1 s, whereas the engine recovery time was less than 3 s after the ordered mode switching. The maximum engine speed and load drops from their initial values were approximately 5% and 4%, respectively. The gas fuel was cut within 1 s in the consecutive firing cylinders with a simultaneous fast increase of the injected diesel fuel, which exhibited an overshoot (obtaining its maximum value at the 11th s of the simulation run) and a subsequent gradual reduction until it reached its steady state value at the 14th s of the simulation run. This is attributed to the PID diesel fuel governor model that detected and appropriately responded to an increased error between the ordered and the actual speed (due to the engine speed drop). Based on the above, it can be concluded that the engine operation complies with the manufacturer specifications/requirements shown in Table 2, according to which the mode switch must occur within 1 s, the acceptable maximum speed drop must be less than 10% and the acceptable maximum recovery time must be 5 s.

It can be observed from Figure 7 that a considerable reduction in the exhaust gas temperature before TC turbine occurred immediately after the gas fuel cut off between the 10.5th s and the 11th s of the simulation run. This is attributed to the fact that the gas fuel was immediately cut off, whilst the diesel fuel rack position response was not as fast (i.e., the diesel fuel cannot instantly reach its required value). This, in turn, resulted in the under-powering of a number of engine cylinders for a number

of engine cycles after the gas full cut off at the 10.5th s associated with a temporary considerable increase of the air–fuel equivalence ratio between the 10.5th s and 11th s of the simulation run, as well as the temporary loss of the engine power, which reached its minimum value at the 11th s of the simulation run. The gradually increasing injection of the diesel fuel (following the gas fuel cutting off) resulted in the recovery of the engine power within 1 s after the time of its minimum value (the engine load almost reached its steady state value at the 12th s of the simulation run), however it caused a notable decrease of the air–fuel equivalence ratio to 1.5 at the 11.5th s. The latter is attributed to the fact that the WG valve was open in the gas mode operation, and therefore the air mass flow rate was less that the one required for the engine to operate in the diesel mode. The lower exhaust gas energy at the TC turbine resulted in corresponding reductions of the TC speed and the boost pressure, which in turn moved the compressor operating point closer to the surge line of the compressor map. For this investigated case, compressor surge did not occur, as the surge margin was adequate; however careful consideration is required during the engine–turbocharger matching procedure to account for the fast-transient phenomena taking place during the GTD mode switching. It must be noted that the predicted temporary engine boost pressure drop is also observed in the experimental results [52] indicating that the simulation effectively captures this feature of the engine operation during the GTD mode switching.

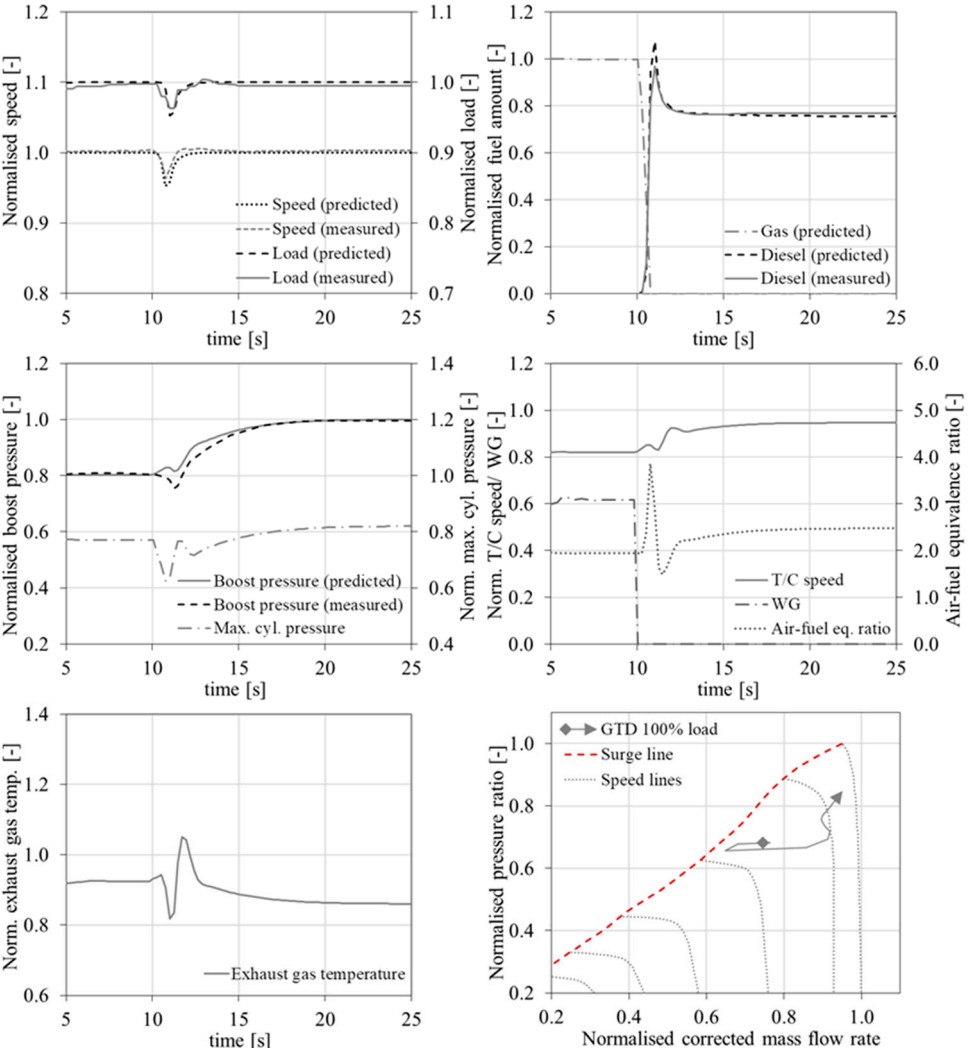

**Figure 7.** Case 1—GTD mode switching at 100% load; predicted engine parameters and comparison with experimental data taken from [52].

The fast increase of the injected diesel fuel in conjunction with the air−fuel equivalence ratio drop resulted in a peak of the exhaust gas temperature (due to the diesel combustion with less air). Following the mode switching order, the engine control system reacted by closing the WG valve, thus increasing the exhaust gas flow rate (hence the energy rate) entering the TC turbine, which in turn increased the TC shaft speed. The gradual increase in the TC shaft speed resulted in a respective increase of the engine boost pressure (hence the engine air flow), which as a consequence gradually increased the engine air−fuel equivalence ratio and reduced the exhaust gas temperature before the TC turbine. All the engine performance parameters reached their steady state values approximately 8 s after the mode switching order; therefore, the engine restored its operation at steady conditions in the diesel mode in the 18th s of the simulation run.

Based on the above analysis, the following points must be noted: (a) a GTD mode switching takes place in a very short space of time (within 1 s) and therefore it is quite challenging for the engine and its control system; (b) the WG valve control along with the engine–turbocharger matching are critical parameters for the successful completion of the GTD mode switching as they affect the compressor normal operation and the turbocharger response time (compressor surging as well as incomplete combustion at the diesel fuel operation must be avoided) with implications to the engine air flow rate, the combustion conditions, and the engine thermal loading; (c) considering that gas fuel operation takes place only for the cylinders initially operating in either the closed cycle or gas injection phase, knocking or misfiring do not seem as an issue during the GTD mode switching.

### 4.2. Case 2—80% Load-DTG Mode Switching

For the second investigated case where the engine operates at 80% load, it can be observed from the respective plots of Figure 8 that the engine speed and load are also predicted with sufficient accuracy; the maximum observed error between the predicted and measured results is less than 2%. However, fluctuations are observed both in the engine speed and load from the experimental measurements, which are attributed to the more considerable cycle to cycle variations of the engine gas mode operation. A notable deviation in the prediction of the diesel fuel amount during this mode switching is observed; however, this can be justified based on the employed method for the gas fuel amount estimation in the modelled engine control system. In the actual engine control system, the gas pressure and the gas valve opening duration are controlled; however, in the model only the latter is controlled, whereas the gas pressure profile is assumed to linearly vary with engine load. In this respect, the fuel transition is gradually performed within 2 min (approximately 100 s) where the gas, diesel and pilot fuels are controlled. Based on the above, it can be concluded that the engine operation complies with the manufacturer specifications shown in Table 2, according to which, the mode switching must occur within 2 min, the acceptable maximum speed drop should be less than 10% and the acceptable maximum recovery time should be 10 s.

In this case, the DTG mode switching is completed within 2 min and as a result, the transition from the diesel engine mode to the gas mode is much slower and smoother compared with the GTD mode switching. In the diesel mode, the engine operates with closed the WG valve, which results in greater values of the boost pressure, the TC shaft speed, as well as the air−fuel equivalence ratio and lower exhaust gas temperature before the TC turbine in comparison with the respective values of these parameters at the gas operating mode. Following the mode switching order, the engine control system reacted by increasing the WG valve opening to its maximum value (considering a WG valve opening limiter in order to avoid compressor surging) at the 11th s of the simulation run; subsequently, the WG valve opening was gradually reduced reaching its steady state value at the end of mode switching (at the 112th s of the simulation run). The WG valve opening resulted in a decrease of the TC shaft speed, and as a consequence, the boost pressure drop, as also can be observed in the experimental results reported in [52]. In turn, this reduced the air flow into the engine cylinders, thus decreasing the air−fuel equivalence ratio whilst increasing the exhaust gas temperature before the TC turbine. The challenges for the engine operation during the DTG mode switching include the knocking

condition avoidance due to the change of the air−fuel equivalence ratio as well as the compressor surging avoidance. Therefore, it can be inferred that the WG valve control is critical for the smooth engine DTG fuel switching.

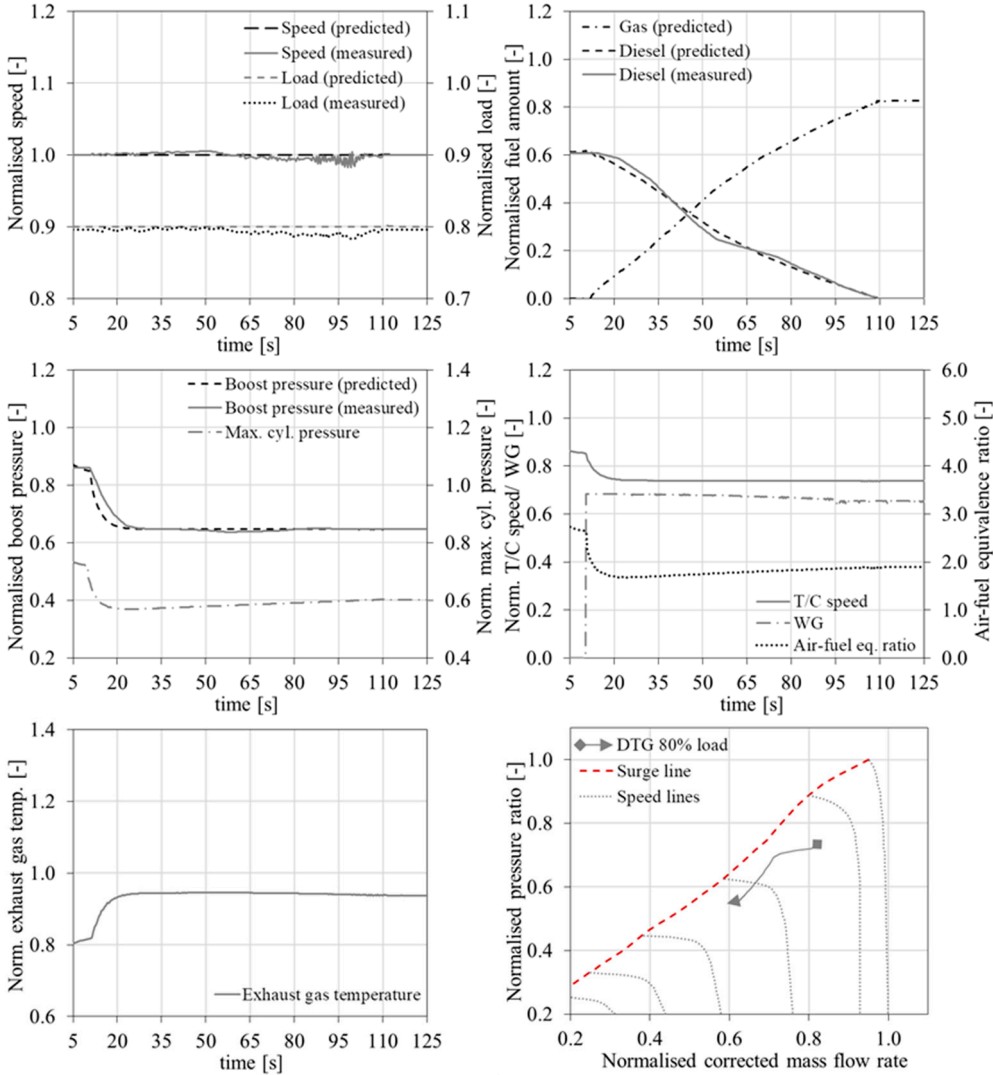

**Figure 8.** Case 2—DTG mode switching at 80% load; predicted engine parameters and comparison with experimental data taken from [52].

### 4.3. Case 3—Step Load Increase from 40% to 80%

For the third investigated case, the engine parameter response is examined under a rather abnormal step-wise load increase from 40% to 80% in the gas mode. As this ordered load change is not permitted in the gas mode according to the engine manufacturer (the maximum allowed step-wise load increase is 20% for the case where the engine operates at 40% load in the gas mode as shown in Figure 2), the engine control system orders a fuel transition from gas to diesel. The measured parameters were taken from [52], where it is reported that they acquired from a plant with two generator sets initially operating at 40% load. One of these two units exhibited an emergency shutdown, so that all the electric load was transferred to the generator set in operation, thus resulting in its almost instantaneous load increase. These engines are of the same type as the investigated engine in this study, however the number of cylinders and their power are double the respective values of the investigated engine herein. Thus, a greater inertia and a relatively slower response is expected in the measured data, as also verified by the results presented in Figure 9. However, as [52] is the only available study in open

literature with published measured results of such a considerable load increase that induces a GTD mode switching, it was chosen to be used for validating the developed model herein.

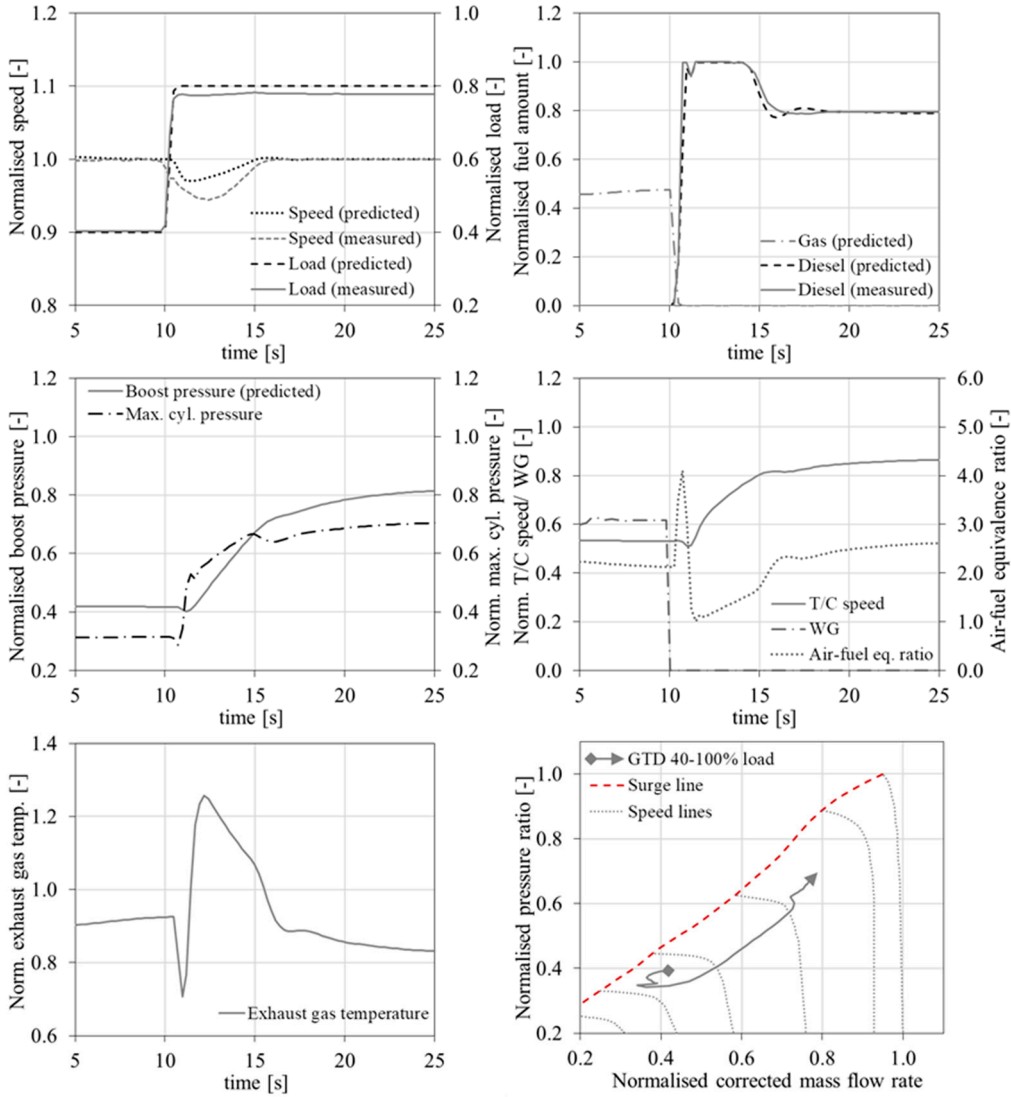

**Figure 9.** Case 3—40–80% load change in the gas mode including a GTD mode switching; predicted engine parameters and comparison with experimental data taken from [25].

The comparison of the derived engine parameter response with the respective experimentally measured variations, which is illustrated in Figure 9, demonstrates that the model can adequately predict the engine response during this transient operation. In specific, the results presented in Figure 9 show that the mode switching from gas to diesel took place within 1 s, whereas the engine recovery time was found to be slightly higher than 5 s. The maximum engine speed drop was found to be 4% and 6% in the simulated and experimental cases, respectively. However, the engine speed response is sufficiently captured. Similarly, to the first investigated case, the gas fuel was cut within 1 s with a simultaneous rapid increase of the injected diesel fuel that retains its maximum value for approximately 3 s due to the PID diesel fuel governor response following the detected increased error between the ordered and the actual speed. The diesel fuel amount started reducing from the 14th s of the simulation run obtaining its steady state value corresponding to 80% load almost after 2 s. An additional characteristic of this case (in comparison with the first investigated case) is the activation of the lambda limiter to confine the injected diesel fuel for values of air−fuel equivalence ratio lower than 1.1. This resulted in the slight drop of the injected diesel fuel amount between the 10.7th and 11th

s of the simulation run. Based on the above discussion, it can be concluded that the engine response complies with the engine manufacturers specifications as described in Table 2.

Similarities of the plotted engine parameters variations with the ones presented and discussed for the first investigated case were observed. The rapid reduction of the gas fuel along with the slower increase of the injected diesel fuel resulted in a temporary considerable increase of the air–fuel equivalence ratio (slightly above 3) as well as a considerable decrease of the exhaust gas temperature before the TC turbine and the corresponding decreases of the TC shaft speed and the boost pressure. Due to the almost instantaneous increase of the engine load and the respective increase of the injected diesel fuel amount, the air–fuel equivalence ratio reduced to very low values close to 1 at the 11th s of the simulation run, denoting that the engine operation temporarily reached the limits for incomplete combustion and smoke appearance. As explained above, the lambda limiter was slightly engaged, reducing the injected diesel fuel amount. The air–fuel equivalence ratio increased after the 11th s of the simulation run as a result of the effect of the WG valve closing and the increase of the exhaust gas temperature (due to the diesel fuel combustion), which resulted in the increase of the exhaust gas energy in the TC turbine and the corresponding increases of the TC shaft speed and the boost pressure (also leading to the reduction of the exhaust gas temperature after the 12th s of the simulation run). The engine speed was restored to its original value at the 15th s of the simulation run, which in turn caused the decrease of the injected fuel to its steady state value and the corresponding changes of the engine parameters slopes variations. The engine operation obtained steady state conditions around the 25th s (7 s longer than what was required for the first investigated case). Similarly to the first investigated case, this run is also quite demanding for both the fuels control system (due to the fast mode switching) and the smooth compressor operation (compressor surging avoidance).

### 4.4. Engine Operational Limitations

In order to identify and map the engine operational limitations for the three investigated cases, a number of engine performance parameters were considered along with their limits (upper or lower) as proposed by the engine manufacturer [53]. These parameters were the following: engine speed, air–fuel ratio equivalence ratio (lambda), exhaust gas temperature before the TC turbine, maximum cylinders WG valve opening/closing area change rate. The employed metrics to characterise the criticality of the engine operation include the percentage difference of each parameter from the respective limit and are provided in Table 3.

From Table 3 results, it can be inferred that, in all the investigated cases, the engine speed drops were within the allowed limits ranges, the cylinders maximum pressure was below its upper limit, whereas the TC shaft speed was kept below its maximum allowed value. On the other hand, the exhaust gas temperature exceeded the respective upper alarm limit for a very limited time for the investigated cases with the GTD mode switching, however the engine shut down limit conditions (alarm limit and duration) were not exceeded. Specifically, for the GTD mode switching in 100% engine load (Case 1) the exhaust gas temperature (before the TC turbine) exceeded the manufacturer limit for 3 s with its maximum value being 5% above the limit. In Case 3 (GTD mode switching and load increase from 40 to 80%), the exhaust gas temperature exceeded the limit for 4.5 s with its maximum value being above the limit value by 26%. In the GTD mode switching, the air–fuel equivalence ratio in the diesel mode exceeded the corresponding lower limit; for Case 1, the lambda exceeded the respective smoke limit for less than 1 s with its minimum value being 6% below this limit; in Case 3, the lambda exceeded the respective smoke (lower) limit for approximately 3 s with its minimum value being 27% below this limit. It should be noted that the usage of a WG valve controller with a slower response or stricter lambda limiter for the diesel fuel can be considered an option to mitigate this operational issue; however, greater speed and power drops are expected in these cases. For the DTG mode switching, the lower air–fuel ratio limit for the gas mode (knocking limit) was critically approached with the lambda value being 6% above this limit. The compressor surge line was also approached for all the investigated cases as shown in Figures 7–9 and the surge margin values presented in Table 3. The most critical cases

for compressor surging were Cases 1 and 2, as the engine operates at high loads. This is connected with the WG valve area change rate, which controls how quickly the WG valve opens or closes. In this respect, it can be inferred that the WG valve almost instant closing is needed for avoiding compressor surging issues and lambda values below the smoke limit for the GTD mode switching, whereas for the DTG mode switching, a slower WG valve opening is required, so that compressor surging and lambda mismatch (leading to knocking or misfiring) are avoided. In cases where the WG valve control fails to provide the specified opening/closing area rates, compressor surging is highly likely to occur.

**Table 3.** Engine operating parameters deviations compared to the respective manufacturer limits.

|  | Case 1 | Case 2 | Case 3 |
| --- | --- | --- | --- |
| **Engine speed** | Within limits; minimum value 5.3% above the lower limit | Within limits; minimum value 9.9% above the lower limit | Within limits; minimum value 7% above the lower limit |
| **Lambda** | For the diesel mode: below the lower limit for less than 1 s; minimum value 6% below the smoke (lower) limit | For the gas mode: minimum value 6% above the knocking (lower) limit | For the diesel mode: below the lower limit for 3 s; minimum value 27% below the smoke (lower) limit |
| **Exhaust Gas temperature before TC** | Above the upper limit for 3 s; maximum value 5% above the upper limit; engine shut down limit was not exceeded | Within limits; maximum value 5% below the upper limit | Above the upper limit for 4.5 s; maximum value 26% above the upper limit; engine shut down limit was not exceeded |
| **Maximum cylinder pressure** | Within limits; Maximum value 18% below the upper limit | Within limits; Maximum value 27% below the upper limit | Within limits; Maximum value 29% below the upper limit |
| **TC speed** | Within limits; Maximum value 5% below the upper limit | Within limits; Maximum value 14% below the upper limit | Within limits; Maximum value 13% below the upper limit |
| **Compressor surge margin** | Minimum value: 5.8% Surge did not occur | Minimum value: 10.5% Surge did not occur | Minimum value: 15.2% Surge did not occur |

The identification of these limitation metrics is quite useful to investigate solutions for mitigating the potential engine safety and operational implications by considering both design measures and engine settings optimisation. Examples of such solutions, which have been adapted by the engine manufacturer in recent engine versions for reducing the likelihood for compressor surging, include the modification of the engine manifolds design to introduce an air bypass loop and a bypass valve for controlling the air flow from the compressor outlet to the turbine inlet [54], as well as the replacement of the exhaust gas WG valve electric actuators by fast acting hydraulic actuators [55].

## 5. Conclusions

The integrated modelling (including the engine thermodynamic model and the control system functional model) of a large marine DF four-stroke engine and its control system was developed in GT-ISE to allow the simulation of the engine transient operation with fuel switching and load changes. The engine control system and component functions were analysed based on the engine manufacturer requirements, thus allowing for the development of the ECS model (also in GT-ISE) to adjust the employed fuels (gas, diesel and pilot) and the exhaust gas WG valve under both steady state and transient operating conditions. The developed integrated model was used for the investigation of three representative engine operating cases with fuel and load changes for which experimental results were published. Based on the analysis of the results, the developed model was validated, the engine and its control system operation were delineated, and the engine operation limitations were discussed taking into consideration the difference in the engine performance parameters from the manufacturer alarm limits. The main findings of the conducted research are summarised as follows.

The model extension/upgrading to accommodate the simulation of the transient conditions with mode switching included: (a) the development of the control functional system model along with the exhaust waste gate control and the fuels control modules; (b) the database development for storing the Wiebe functions parameters as well as the combustion modelling during the various modes switching, and; (c) addition of the transient simulations input parameters.

The challenge for the control system functional modelling was mainly attributed the complexity of this system as well as the limited information available in literature.

The combustion modelling was challenging, as the model needed to accommodate the combustion process of the employed engine fuels (gas, diesel and pilot) during the mode switching.

The followed approach for predicting the engine combustion performance during transients, which included a database for storing the Wiebe functions parameters validated at steady state conditions and the use of quadratic interpolation for predicting the combustion characteristics during the fuel and load changes, proved to be sufficient for capturing the involved phenomena.

The developed engine control system represented the actual ECS operation realistically. However, it was a demanding task, requiring attention to all the engine control subsystems and components as well as their interactions, the sequence of the involved processes and the engine manufacturer requirements.

From the performed validation study, it was inferred that the developed model is capable of sufficiently predicting the engine parameters response during transient operating conditions, including fuel and load changes. Even for the almost instantaneous GTD mode switching, in which modelling and simulation are fairly demanding tasks, the developed model succeeded in providing sufficient predictions.

The developed tool provides the required accuracy and can be used with fidelity for investigating both the steady state and transient engine operation.

The engine response for the investigated operating cases were found to satisfy the engine manufacturer requirements.

The mode switching from gas to diesel (GTD) is rapid and must be completed within 1 s; therefore, it is demanding for both the engine and its control system operation. The quick gas fuel cut off in conjunction with the slower diesel fuel increase resulted in a temporary loss of power, associated with a slight reduction in the TC shaft speed and the engine boost pressure. The subsequent reduction of the air−fuel equivalence ratio and the increase of the exhaust gas temperature can result in smoke and engine component thermal loading, respectively.

The mode switching from diesel to gas (DTG) has to be completed in a longer period (within 2 min), and as a result, smoother engine parameters variations were observed. The WG valve control is critical for avoiding compressor surging issues as well as air−fuel equivalence ratio mismatching between the diesel and the gas operating modes.

The exhaust gas WG valve control is a crucial part of the sophisticated and complex ECS for ensuring the smooth and reliable marine DF engine transient operation with fuel and changes.

Careful consideration is required during the engine–turbocharger matching and the engine control system design to accommodate the contradictory requirements for the GTD and DTG fuel changes.

The developed model's computational time on a modern desktop computer with a single processing unit is approximately 20–25 times the real time. The simulation time can be significantly reduced by using a parallel computing approach. It must also be noted that the GT-ISE offers a tool to develop a real time model, which can be used in future work for developing the real time digital twin of the investigated engine.

In conclusion, the results of the derived analysis revealed the engine's and its systems' characteristics during transients with fuel and load changes, enlightening the involved processes and associated phenomena. In this respect, the developed model will be a useful tool for further developing, testing and designing engine control system components. This model could form the basis to develop an engine real-time digital twin, which can be combined with the engine monitoring system and machine learning algorithms to provide new directions that ensure smooth, reliable and efficient engine operation.

**Author Contributions:** Conceptualization, S.S., G.T.; methodology, S.S., G.T., C.M.; software, S.S., C.M.; validation, S.S., G.T., C.M.; formal analysis, S.S., G.T., C.M.; writing—original draft preparation, S.S., G.T., C.M.; writing—review and editing, G.T., E.B.; visualization, S.S.; supervision, G.T., E.B.; project administration, G.T. All authors have read and agreed to the published version of the manuscript.

**Funding:** No funding was received for this research.

**Acknowledgments:** The authors greatly acknowledge the funding from DNV GL AS and RCCL for the MSRC establishment and operation. The opinions expressed herein are those of the authors and should not be construed to reflect the views of DNV GL AS and RCCL. Gamma Technologies support is also greatly acknowledged by the authors.

**Conflicts of Interest:** No potential conflict of interest was reported by the authors.

## Nomenclature List

| | |
|---|---|
| A | Pipe cross-sectional flow area ($m^2$) |
| BMEP | Brake Mean Effective Pressure (bar) |
| BSEC | Brake Specific Energy Consumption (kJ/kWh) |
| BSFC | Brake Specific Fuel Consumption (g/kWh) |
| $C_f$ | Friction factor (−) |
| D | Pipe diameter (m) |
| dp | Pressure differential acting across dx (Pa) |
| dx | Discretization length (m) |
| $E_{f,total}$ | Total fuel energy (J) |
| e | Total specific internal energy (J/kg) |
| FF | Weight of Wiebe function (−) |
| H | Total specific enthalpy (J/kg) |
| $K_p$ | Pressure loss coefficient (−) |
| LHV | Lower heating value (J/kg) |
| m | Mass (kg); Wiebe function parameter m (−) |
| $\dot{m}$ | Mass flow rate (kg/s) |
| $m_d$ | Diesel fuel mass (kg) |
| $m_f$ | Burnt fuel amount (kg) |
| $m_g$ | Gas fuel mass (kg) |
| $m_p$ | Pilot fuel mass (kg) |
| p | Pressure (Pa) |
| $\dot{Q}_{ht}$ | Heat flow rate (W) |
| $\dot{Q}_b$ | Combustion heat release rate (W) |
| u | Velocity (m/s) |
| V | Volume ($m^3$) |
| Greeks | |
| $a$ | Wiebe function parameter (−) |
| $\dot{x}_b$ | Fuel burning rate (−) |
| $\Delta\theta_{g,i}$ | Combustion duration (deg) |
| θ | Crank angle (deg) |
| $\theta_{SC\_i}$ | Start of combustion (deg) |
| λ | Air−fuel equivalence ratio (−) |
| ρ | Density ($kg/m^3$) |

## Abbreviation List

| | |
|---|---|
| 0D | Zero-dimensional |
| 1D | One-dimensional |
| 3D | Three-dimensional |
| BMEP | Brake Mean Effective Pressure |
| BSEC | Brake Specific Energy Consumption |
| BSFC | Brake Specific Fuel Consumption |

| CA | crank angle |
| CO | Carbon Monoxide |
| $CO_2$ | Carbon Dioxide |
| DF | Dual Fuel |
| DTG | Diesel to Gas fuel change |
| ECA | Emission Control Area |
| ECS | Engine Control System |
| EEDI | Energy Efficiency Design Index |
| EEOI | Energy Efficiency Operational Indicator |
| GTD | Gas to Diesel fuel change |
| HC | Hydrocarbons |
| HCCI | Homogeneous Charge Compression Ignition |
| HFO | Heavy Fuel Oil |
| IMO | International Maritime Organization |
| LHV | Lower Heating Value |
| LNG | Liquefied Natural Gas |
| MARPOL | International Convention for the Prevention of Marine Pollution |
| MCR | Maximum continuous rating |
| MGO | Marine Gas Oil |
| MISO | Multiple Input Single Output |
| NG | Natural Gas |
| NOx | Nitrogen Oxides |
| PID | Proportional–Integral–Derivative controller |
| PM | Particulate Matter |
| SOx | Sulphur Oxides |
| TC | Turbocharger |
| WG | Exhaust gas waste gate |

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
