# Peer review of "Towards Marine Dual Fuel Engines Digital Twins—Integrated Modelling of Thermodynamic Processes and Control System Functions"

_jmse, doi:10.3390/jmse8030200_

Round 1

Reviewer 1 Report

This is a very well written paper, highlighting the capabilities of a computational model to serve as a digitial twin for an engine. I have some minor comments / suggestions for the authors: 

  1. In the references section the authors bring up the point that very few studies have focussed on modeling marine dual fuel engines. I would like to point the authors to another relevant and recent reference - "Towards Predictive Dual-Fuel Combustion and Pre-Chamber Modeling for Large Two-Stroke Engines in the Scope of 0D/1D Simulation", CIMAC World Congress, Vancouver 2019. 
  2. Was the a reason the authors decided to adopt a triple wiebe function to represent the combustion rate instead of a more predictive model, which contains the relevant physics needed to respond to transient variations, such as the one used in the paper mentioned above? 
  3. Page 8, line 292 - the authors mention "gas fuel injectors". However the model shown in figure 3 does not show any gas fuel injector components. Was the gas fuel injection modeled using an injector or was it modeled using a valve? 
  4. Page 11, line 399 - "... the following three controlling parameters" - only 2 parameters are mentioned.
  5. In the section starting from line 401 the authors refer to Figure 5. However I think the reference should be to figure 4. 
  6. Line 408 - " ... (single Wiebe function model for the diesel mode)" - Was a single Wiebe sufficient to capture both the premixed and diffusion burns typically seen in diesel combustion?
  7. In a simulation setting, it is easy to ensure that the switching of combustion mode occurs only when the cylinder is in the right angle range, i.e. combustion has not already started in a particular mode. For the sake of my curiosity, how is this ensured in an actual engine control system. Does the control system have a way to track the local angle (position) each cylinder is at? 
  8. Equation 7 - what is E_f,total?
  9. Figure 5 - what is CRM?
  10. Line 512 - what does "open cycle phase" refer to?
  11. Figures 7, 8, 9 - spelling of "equivalence" needs to be fixed.
  12. Figures 7, 8, 9 - how is the air equivalence ratio calculated?
  13. Line 662 - The maximum allowed step load increase in diesel mode is less than 35%. How was a step load increase of 40% allowed?
  14. What was the runtime (factor of RT) of this compuational model? 

Author Response

Response to Reviewers’ comments letter

The authors would like to thank the reviewers for the detailed and insightful comments. We made every possible effort to address these comments and improve the original manuscript. Please kindly find below our responses in red.

Reviewer 1

This is a very well written paper, highlighting the capabilities of a computational model to serve as a digital twin for an engine. I have some minor comments / suggestions for the authors:

  1. In the references section the authors bring up the point that very few studies have focussed on modelling marine dual fuel engines. I would like to point the authors to another relevant and recent reference - "Towards Predictive Dual-Fuel Combustion and Pre-Chamber Modelling for Large Two-Stroke Engines in the Scope of 0D/1D Simulation", CIMAC World Congress, Vancouver 2019.

The proposed reference was added in the manuscript. Slight amendments were made in the Introduction section to accommodate this reference.

  1. Was there a reason the authors decided to adopt a triple Wiebe function to represent the combustion rate instead of a more predictive model, which contains the relevant physics needed to respond to transient variations, such as the one used in the paper mentioned above?

A predictive combustion model requires a considerable set of the model constant calibration. Although the authors had access to the engine test trials, data to calibrate a predictive combustion model was not available. On the other hand, triple Wiebe function models have been used to represent DF engines combustion providing adequate accuracy. A comment was added in Section 3.2 (lines 402-417).

  1. Page 8, line 292 - the authors mention "gas fuel injectors". However, the model shown in figure 3 does not show any gas fuel injector components. Was the gas fuel injection modelled using an injector or was it modelled using a valve?

This was an oversight from our side. The engine includes electronically controlled solenoid gas admission valves. The manuscript was accordingly replacing “gas injectors” with “gas admission valves”.

  1. Page 11, line 399 - "... the following three controlling parameters" - only 2 parameters are mentioned.

Indeed, the controlling parameters are two. The manuscript was revised accordingly.

  1. In the section starting from line 401 the authors refer to Figure 5. However, I think the reference should be to figure 4.

Indeed that was the case. The manuscript was revised accordingly.

  1. Line 408 - " ... (single Wiebe function model for the diesel mode)" - Was a single Wiebe sufficient to capture both the premixed and diffusion burns typically seen in diesel combustion?

The single Wiebe function can sufficiently capture the combustion phenomena in the diesel mode. For this study, the cylinder pressure diagram was not available. However, by using the single Wiebe combustion model, the maximum cylinders pressure, the Brake Specific Fuel Consumption (BSFC) and the Indicated Mean Effective Pressure (IMEP) for the engine steady state operation were predicted with adequate accuracy. This is commented in Section 3.2 (lines 402-417).

  1. In a simulation setting, it is easy to ensure that the switching of combustion mode occurs only when the cylinder is in the right angle range, i.e. combustion has not already started in a particular mode. For the sake of my curiosity, how is this ensured in an actual engine control system. Does the control system have a way to track the local angle (position) each cylinder is at?

It must be noted that in the ECS model, the CA position information was taken from the engine crankshaft block whist considering the phase angle of each cylinder. The actual engine control system employs two crank angle (CA) sensors (encoders) for measuring the crank angle; the first is located in the flywheel whereas the second is located in the free end.

The preceding paragraph was added as the last paragraph of Section 3.3.

However, as it is illustrated from Fig. 5 and 6, the control functions are quite complicated. In this study, the functional control system model was developed to represent (as closer as possible) the actual engine control system.

  1. Equation 7 - what is E_f,total?

Ef,total is the total energy of all the injected fuels and it was introduced following eq. (3). E­q­ in the original manuscript was a typo, which is now corrected in the revised manuscript.

  1. Figure 5 - what is CRM?

CRM denoted the Current Running operating Mode. The description of this abbreviation was added to Fig. 5 caption.

  1. Line 512 - what does "open cycle phase" refer to?

It refers to the open cycle. To avoid confusion, this sentence was slightly modified to “… the ECS model allows for the gas fuel injection only into for the cylinders operating in their open cycle period".

  1. Figures 7, 8, 9 - spelling of "equivalence" needs to be fixed.

These figures were revised accordingly.

  1. Figures 7, 8, 9 - how is the air equivalence ratio calculated?

The equivalence ratio is provided as output from the GT Power. Although we tried to find the formula from its calculation from GT-Power manuals, it seems that his information was not provided. Therefore, we wouldn’t like to include a statement in the revised manuscript that cannot be verified. I hope the reviewer understands this.

  1. Line 662 - The maximum allowed step load increase in diesel mode is less than 35%. How was a step load increase of 40% allowed?

The specific case elaborates a scenario where the engine operates at 40% load in the gas mode and the ordered load increase is higher than the allowed step load increase (which is 20%). Therefore, a mode switching (from the gas to diesel mode) is required in this case.

This scenario does not lie under the given transient operational limitations of Table 2.

  1. What was the runtime (factor of RT) of this computational model?

On a modern desktop computer, the simulation time when a single processor is used (due to licence limitations) is approximately 20-25 times the real time. The simulation time can be significantly reduced by using parallel computing approach. It must also be noted that the GT-ISE offers a tool to develop a real time model, which can be used in future work for developing a real time digital twin. A comment was added in the conclusions section.

Reviewer 2 Report

This is a very interesting study that provides valuable information on a highly demanding and nowadays challenging subject related to DF engine operations and specifically the systematic investigation of a marine four-stroke dual-fuel engine response during transient operation with fuel modes switching and load changes.

In general, the study is adequately presented with detailed information regarding the procedure followed and extensive discussion on the results. Therefore this paper is highly recommended for publications (accepted as is).

However, please consider the following comments/remarks and it would be beneficial for the impact of the study if you provided the information/explanations/corrections suggested in the final version of the paper.

1) In the text, you describe the model used for predicting emissions (NOx) however, in the study presented no such information/results have been provided. Therefore, it is suggested that the part of the model description corresponding to the emission to be omitted (or if you prefer it would be more than welcomed if you added a section with the corresponding results of emissions, under the cases examined)

2) It is clearly mentioned in the text that the formulation of the basic version of the model has been presented in another publication. However, for the completeness of the paper, it would be useful for the interested researcher to provide the values of the constants used referred in lines 355 and 356 (specifically heat transfer coefficient, heat transfer area and cooling water temperature)

3) Please make a comment if the cooling water temperature variation is taken into account during the transient modes examined.

4) Based on the text lines 365-370, please explain why on pure diesel mode a single Wiebe function is accounted to be adequate for describing the combustion process, while on DF mode with pilot injection a dual Wiebe function is needed (for premixed and diffusive phase). It seems logical since when injecting a very low amount of fuel the relative importance of these two phases is increased, but based on your experience is this increased complexity on modeling necessary?

5) In line 379, please explain how the weight of each Wiebe function is determined.

6) In line 393, it would be useful to provide more information regarding the calibration process and specifically how do you determine the constants used in 3 different combustion mechanisms using an integrated (experimental) combustion rate.

7) Line 564, Line 624 and Line 660, it is suggested to provide a short description (text) of the case examined (for the reader to have a better overview)

8) Line 756, please correct word "resent" with "recent"

9) Please make a short comment on the possible increased complexity/problems that should be overcome in the model formulation, if engine speed transient modes would be of interest to be examined.

10) It would be interesting to provide some information regarding the required computational time of the model (important parameter for the digital twin concept).

11) Please accept a suggestion for a future publication. It would be of great interest if the authors could present results regarding the exhaust gas emission under transient operation, as well as present cases with transient operation in the gas mode which does not lead to fuel switch operation (diesel mode)

Concluding, this is a very interesting study that is based on a very "powerful" engine model that can provide insight into complex phenomena that are crucial for engine operation and control and give answers to real-world engineering problems.

Author Response

Response to Reviewers’ comments letter

The authors would like to thank the reviewers for the detailed and insightful comments. We made every possible effort to address these comments and improve the original manuscript. Please kindly find below our responses in red.

Reviewer 2

This is a very interesting study that provides valuable information on a highly demanding and nowadays challenging subject related to DF engine operations and specifically the systematic investigation of a marine four-stroke dual-fuel engine response during transient operation with fuel modes switching and load changes. In general, the study is adequately presented with detailed information regarding the procedure followed and extensive discussion on the results. Therefore, this paper is highly recommended for publications (accepted as is).

However, please consider the following comments/remarks and it would be beneficial for the impact of the study if you provided the information/explanations/corrections suggested in the final version of the paper.

  1. In the text, you describe the model used for predicting emissions (NOx) however, in the study presented no such information/results have been provided. Therefore, it is suggested that the part of the model description corresponding to the emission to be omitted (or if you prefer it would be more than welcomed if you added a section with the corresponding results of emissions, under the cases examined)

The predicted CO2 and NOx emissions for the first two investigated cases (Cases 1 and 2) are presented in:

Theotokatos, G, Stoumpos, S, Bolbot, V, & Boulougouris, E. (2020), Simulation-based investigation of a marine dual-fuel engine, Journal of Marine Engineering & Technology, 19:sup1, 5-16, DOI: 10.1080/20464177.2020.1717266.

However, for the completeness of in the present study, the authors included information on the emissions prediction model.

  1. It is clearly mentioned in the text that the formulation of the basic version of the model has been presented in another publication. However, for the completeness of the paper, it would be useful for the interested researcher to provide the values of the constants used referred in lines 355 and 356 (specifically heat transfer coefficient, heat transfer area and cooling water temperature).

Due to the fact that this is one of the few available simulation studies considering the transient operation of DF engines with modes switching, the authors expect that not only the air cooler and heat transfer model constants but also the other submodels constant will be on interest for various researchers. In this respect, the authors are willing to share this information upon private communication with the interested researchers.

  1. Please make a comment if the cooling water temperature variation is taken into account during the transient modes examined.

In this study, the cooling water temperature is considered constant for simulating both steady state and transient conditions. A comment was added (line 362-363) to clarify this point.

  1. Based on the text lines 365-370, please explain why on pure diesel mode a single Wiebe function is accounted to be adequate for describing the combustion process, while on DF mode with pilot injection a dual Wiebe function is needed (for premixed and diffusive phase). It seems logical since when injecting a very low amount of fuel the relative importance of these two phases is increased, but based on your experience is this increased complexity on modelling necessary?

The single Wiebe function can sufficiently capture the combustion phenomena in the diesel mode. For this study, the cylinder pressure diagram was not available. However, by using the single Wiebe combustion model, the maximum cylinders pressure, the Brake Specific Fuel Consumption (BSFC) and the Indicated Mean Effective Pressure (IMEP) for the engine steady state operation were predicted with adequate accuracy. For the case of gas mode operation, it was only the triple Wiebe function model that provided sufficient accuracy for these parameters. This is commented in Section 3.2 (lines 402-417).

  1. In line 379, please explain how the weight of each Wiebe function is determined.

The weighting factors of each Wiebe function were calibrated, so that the experimentally available parameters (maximum cylinders pressure, the Brake Specific Energy Consumption (BSEC) and the Indicated Mean Effective Pressure (IMEP)) are predicted with sufficient accuracy. This is commented in Section 3.2 (lines 402-417).

  1. In line 393, it would be useful to provide more information regarding the calibration process and specifically how do you determine the constants used in 3 different combustion mechanisms using an integrated (experimental) combustion rate.

To represent the engine operation in the gas mode, the following parameters were calibrated for each Wiebe function: the weighting factors, the combustion duration, the start of combustion and the shape factors. These parameters are calibrated to predict with adequate accuracy the maximum cylinders pressure, the Brake Specific Energy Consumption (BSEC) and the Indicated Mean Effective Pressure (IMEP). This is commented in Section 3.2 (lines 402-417).

  1. Line 564, Line 624 and Line 660, it is suggested to provide a short description (text) of the case examined (for the reader to have a better overview)

The three cases are briefly described earlier in the same section. To provide more clarity, the titles of subsections 4.1, 4.2 and 4.3 were revised.

  1. Line 756, please correct word "resent" with "recent"

Corrected.

  1. Please make a short comment on the possible increased complexity/problems that should be overcome in the model formulation, if engine speed transient modes would be of interest to be examined.

The main issue to develop this model is the unavailability of information and the confidentiality of the available data. It was quite challenging to gather the required material for developing the model. The authors gathered and elaborated various diverse sources of material. Other challenges were the complexity of the control system functions the combustion model. A comment was added in the conclusions section.

  1. It would be interesting to provide some information regarding the required computational time of the model (important parameter for the digital twin concept).

On a modern desktop computer, the simulation time when a single processor is used (due to licence limitations) is approximately 20-25 times the real time. The simulation time can be significantly reduced by using parallel computing approach. It must also be noted that the GT-ISE offers a tool to develop a real time model, which can be used in future work for developing a real time digital twin.

  1. Please accept a suggestion for a future publication. It would be of great interest if the authors could present results regarding the exhaust gas emission under transient operation, as well as present cases with transient operation in the gas mode which does not lead to fuel switch operation (diesel mode).

Indeed this is a good suggestion. As mention above, the predicted CO2 and NOx emissions for the first two investigated cases (Cases 1 and 2) are presented in:

Theotokatos, G, Stoumpos, S, Bolbot, V, & Boulougouris, E. (2020), Simulation-based investigation of a marine dual-fuel engine, Journal of Marine Engineering & Technology, 19:sup1, 5-16, DOI: 10.1080/20464177.2020.1717266.

We have initiated a study to investigate emissions reduction methods in the diesel mode also considering NOx Tier III levels, so it will be included in one of our future studies.